



# Anthropogenic climate change and glacier lake outburst flood risk: local and global drivers and responsibilities for the case of Lake Palcacocha, Peru

Christian Huggel[1], Mark Carey[2], Adam Emmer[3], Holger Frey[1], Noah Walker-Crawford[4], Ivo Wallimann-Helmer[5]

[1]Department of Geography, University of Zurich, Winterthurerstrasse 190, CH-8057 Zurich, Switzerland
[2]Robert D. Clark Honors College and Environmental Studies Program, University of Oregon, USA
[3]The Czech Academy of Science, Global Change Research Institute, 603 00 Brno, Czech Republic
[4]Department of Social Anthropology, Manchester University, UK
[5]Environmental Humanities; Department of Geosciences, University of Fribourg, Switzerland

*Correspondence to*: Christian Huggel (christian.huggel@geo.uzh.ch )

**Abstract.** Evidence of observed negative impacts on natural and human systems from anthropogenic climate change is increasing. However, human systems in particular are dynamic and influenced by multiple drivers, and hence identifying an anthropogenic climate signal is challenging.

Here we analyze the case of lake Palcacocha in the Andes of Peru which offers a representative model for other glacier lakes and related risks around the world because it features a dynamic evolution of flood risk driven by physical and socio-economic factors and processes. Furthermore, it is the object of a prominent climate litigation case where a local Peruvian citizen sued a large German energy producer over risk of flooding from lake Palcacocha.

Adopting a conceptual model of cascading impacts and multiple drivers of risk we first study climatic and other geophysical drivers of flood risk. We find that an anthropogenic signal related to greenhouse gas emissions is traceable. In parallel, flood risk has been strongly shaped (and increased) by interacting socio-economic, institutional and cultural processes over the past decades.

The case raises important questions of responsibility for flood risk of global and local agents which, however, are difficult to address in cases like Palcacocha where we reveal a complex network of interlinked global, national and local drivers. Following from this we outline a normative framework with a differentiated perspective on responsibility, implying that global emitters commit to support strengthening capacities in affected regions and localities, and local institutions and societies engage in local risk reduction measures and policies.

## 1 Introduction

Impacts of climate change are increasingly observed in many natural and human systems worldwide (Cramer et al., 2014; Hoegh - Guldberg et al., 2018). Shrinking glaciers are among the most visible indicators of climate change as the mountain cryosphere is especially prone to warming (Braun et al., 2019; Hock and et al., 2019; Zemp et al., 2015). While glaciers are





widely monitored from the ground and from space, the impacts of glacier changes on natural and human systems are often more difficult to observe, and attribution of the observed changes to causal factors can be challenging (Carey et al., 2017; Hansen and Stone, 2016; Huggel et al., 2016). Changes in water resources and natural hazards are thereby the most substantial

effects, and have been documented in many mountain regions of the world (Casassa et al., 2009; Cramer et al., 2014; Harrison et al., 2018). Glacier lake outburst floods (GLOF) are among the most destructive and far-reaching hazards related to glacier changes and have killed thousands of people in single events (Carey, 2005; Carrivick and Tweed, 2016).

Glaciers will continue to shrink and impact downstream natural and human systems in the coming decades, although emission pathways will have a crucial effect on the extent of the process and impacts (Hock and et al., 2019; Huss and Hock, 2018;

Kraaijenbrink et al., 2017; Schauwecker et al., 2017). Adaptation to cryosphere impacts is of fundamental importance and has taken place so far in the majority of countries (McDowell et al., 2019). However, adaptation to cryosphere change may reach certain limits, e.g. with the disappearance of glaciers in regions highly dependent on glacier melt water, or large slope instabilities making certain areas uninhabitable or existing livelihood strategies unviable, thus resulting in losses and damages (Huggel et al., 2019). Loss and Damage (L&D) as a concept in global climate policy has been defined as the impacts that

cannot or have not been avoided through mitigation and adaptation (Okereke et al., 2014; Warner and van der Geest, 2013), but there is still missing clarity and debate about what L&D comprises (Calliari, 2018; Lees, 2017; Mechler et al., 2019). In the Paris Agreement L&D was anchored in a separate article, but at the same time the agreement specifies that this article does not provide any basis for liability and compensation.

Despite this disclaimer at the level of international policy, important questions of responsibility and justice emerge from

negative effects and risks related to climate change in general and to the mountain cryosphere specifically, such as: which natural and social processes can be identified as drivers of risk; to what extent are global greenhouse gas emitters contributing to these risks; who must be held accountable to reduce local loss of lives and goods; and under what circumstances are local people, institutions and governments able to manage these risks? There is currently only limited research that offers evidence for and responses to these questions. In this paper, we analyze these aspects from different disciplinary perspectives and

associate them with a normative responsibility framework.

This paper focuses on the glacier lake Palcacocha and associated flood risk for the downstream city of Huaraz in the Peruvian Andes (Fig. 1) to help answer these questions in a specific context and to offer larger insights into climate change risks and responsibilities. While being attentive to a diversity of risk frameworks and concepts (Blaikie et al., 1994; Oliver-Smith, 2013;

Wisner et al., 2004), we understand risk as a function of physical hazard, human exposure, and vulnerability of people and assets (IPCC, 2014). Lake Palcacocha offers a representative case for other glacier lakes and related risks around the world because it features many physical and social dynamics found elsewhere: a shrinking glacier that led to the formation of a large glacier lake where ice previously existed; continued lake instability due to glacier retreat and moraine dam instability; a past glacier lake outburst flood that killed thousands of local residents and partially destroyed a city and other communities and

infrastructure; repeated flood prevention and lake drainage engineering works; a history of glacier lake monitoring and ongoing scientific studies; contested knowledge, science and perceptions about the lake and its risks among experts, policymakers, local residents, and other groups; a complex political and institutional context with periods of increased attention and neglect



of the problem by authorities and the local population; unclear responsibilities among different government agencies and levels; and, overall, a dynamic evolution of risk driven by physical and socio-economic factors and processes. Our objective is to analyze to what extent we can identify natural and social processes as factors and drivers of risk at lake Palcacocha and in Huaraz, and discuss whether this analysis can inform the conceptualization of responsibilities related to managing the negative impacts of anthropogenic climate change.

This analysis is timely not only because glacierized mountain regions are increasingly grappling with unstable glacier lakes, but also because Lake Palcacocha recently made headlines worldwide because of a legal case, Saúl Luciano Lliuya vs RWE (Frank et al., 2019). This case, pending at a German court,[1] emerged when a local resident of Huaraz (Luciano Lliuya), sued the German energy producer RWE over flood risk from glacier lake Palcacocha, threatening his property if the lake were to cause a flood. Luciano Lliuya argues that Palcacocha is unstable as a result of anthropogenic emissions (to which, he alleges, RWE made a significant contribution), which caused glacier retreat and the growth of lake Palcacocha, making the lake unstable and threatening downstream communities. The case was initially dismissed at a local German court but then admitted by a higher appeals court. As the first lawsuit of its kind to reach this stage, it was considered a significant breakthrough in climate litigation (Ganguly et al., 2018; Huggel et al., 2016). Although the literature on climate litigation is steadily growing (Marjanac and Patton, 2018; McCormick et al., 2018), questions of responsibility, and possibly liability, in a case like Palcacocha, remain mostly unanswered and hence call for studies that analyze risk and responsibility during climatic, cryospheric, and societal change.

To achieve a comprehensive picture of flood risk in Huaraz and the relation to climate and socio-economic change we make use of existing data and information, and conduct additional research where needed, including hazard field studies, numerical flood modeling, satellite data analysis, census data and interviews. We structure our paper as follows: we first analyze the physical evolution of lake Palcacocha from the mid-nineteenth century Little Ice Age (LIA) to the present (section 2). We then disentangle the different drivers of GLOF risks following the aforementioned IPCC-based risk concept. We begin with the physical hazard component of risk, studying how global drivers of anthropogenic greenhouse gas emissions affect the local conditions of GLOF hazard (section 3). We collected remote sensing and fieldwork-based information to document the evolution of lake Palcacocha. We analyzed the hazard conditions at and around the lake to develop a number of flood scenarios that we then implemented in a physically based GLOF flow model following established methodologies to evaluate the downstream hazard in Huaraz. We then look at how social, economic, institutional, and cultural aspects become drivers of risk exposure and vulnerability (sections 4-6). For this purpose, we used a mixed methods approach to elucidate various environmental and socio-cultural drivers that contribute to risk. Our historical analysis of risk development in Huaraz derives primarily from historical document analysis, literature review and interviews with contemporaneous figures. The more recent analysis of socio-cultural and institutional factors contributing to risk (since 2009) draws on qualitative data from participant observation in institutional, urban and rural settings as well as interviews with people from these three spaces.

---

[1] Saúl Ananías Luciano Lliuya ./. RWE AG, Oberlandesgericht Hamm, Az.: I-5 U 15/17



This case demonstrates the inherent links between local and global activities to manage climate risks, and how they drive localized climate-related risks. The local-global linkages raise many questions about causality, responsibility and justice (section 7). Our assumption is that a better understanding of the diverse drivers of risk in a case like Palcacocha allows us to clarify the differentiation of responsibilities and the challenges ahead vis-à-vis impacts and risks of loss and damage..

## 2 Lake Palcacocha

The evolution and history of Lake Palcacocha in Peru's Cordillera Blanca is linked to glacier retreat, driven by both natural and anthropogenic forcing, large flood disasters as well as human intervention and flood mitigation at the lake (see Tab. 1). According to the lichenometric dating, the moraine which later dammed the lake developed between 1590 and 1630 (Emmer, 2017) by advancing glaciers from the southwestern slopes of Palcaraju (6,274 m a.s.l.) and Pucaranra (6,156 m a.s.l.) mountains. This period corresponds to the beginning of the first, more distinct phase of the LIA in the Cordillera Blanca (Thompson et al., 2000; Solomina et al., 2007). It is not known precisely when the lake formed; however, based on the available evidence we estimate that it was likely after the second phase of the LIA in the second half of 19[th] Century. The Palcacocha drainage outlet nourishes the Paria river, joins other waterways downvalley and flows into the Quillcay River that runs through downtown Huaraz, the capital city of Peru's Ancash Region with approximately 140,000 inhabitants today.

The first scientific expeditions and observations of the lake were undertaken by Austrian and German researchers, led by Hanz Kinzl in the late 1930s, a time before anyone realized the threat that Palcacocha posed to downstream communities (Carey, 2012; Portocarrero, 2014; Wegner, 2014). Shortly after, on 13 December 1941, Lake Palcacocha's moraine dam failed, resulting in a GLOF with a volume of > 10 million $m^3$ and peak discharge in excess of 10,000 $m^3$/s, producing devastating impacts in the city of Huaraz, located ca. 25 km downstream from the lake (Mergili et al., 2020; Somos-Valenzuela et al., 2016). The flood killed nearly 2000 people and destroyed one-third of the city of Huaraz, particularly its most developed downtown area and modern commercial district (Carey, 2010; Wegner, 2014). This event is considered among the worst floods ever documented worldwide resulting from natural dam failure (Carrivick and Tweed, 2016; Costa and Schuster, 1988). Although the precise cause is not known, the 1941 flood likely followed an impact of an ice avalanche into the lake or failure related to piping in the dam (Oppenheim, 1946).

After the 1941 GLOF, a small 'residual' lake remained, dammed by the basal moraine (elevated part of the former bottom of the lake basin). The lake volume remained relatively stable for several decades (Figs. 2, 3), even when a heavy earthquake on 31[st] May 1970 (M=7.9) caused disastrous effects on the region (Lliboutry et al., 1977). In the early 1970s, after almost three decades of minimal lake growth or stagnation, a permanent drainage canal and two artificial dams were constructed, lowering the lake water level by 1 m, and stabilizing it at 4,566 m a.s.l. with 7 m of freeboard and $0.515 \cdot 10^6$ $m^3$ of water (Table 1). The contemporary period of glacier retreat and downwasting accompanied by lake expansion started by the end of the 1970s and extends to present (Vilimek et al., 2005).

On 19[th] March 2003, the left lateral moraine along lake Palcacocha slid into the lake and produced a displacement wave that overtopped the dam and caused a small lake outburst flood further downvalley. This flood, combined with an inaccurate but high-publicity NASA announcement one month later in April 2003 about glacier instability above Palcacocha re-opened



discussions about the lake's threat to the city of Huaraz (Carey, 2010; Kargel et al., 2011). Given the more than 110,000
       inhabitants of Huaraz at the time, these events led to a number of new hazard and risk assessment studies (e.g., (Hegglin and
       Huggel, 2008; Vilimek et al., 2005). The lake volume at that time was determined as $3.690 \cdot 10^6$ m$^3$ (+640% in 29 years; (Zapata
       et al., 2004), although some doubts have arisen concerning the accuracy of this 2003 lake bathymetry and the lake area in fact
       would suggest a higher lake volume. The existing hazard mitigation measures built in the 1970s were no longer found sufficient

(Emmer et al., 2018). In 2009, a new bathymetry revealed that the lake had grown to $17.325 \cdot 10^6$ m$^3$ (Portocarrero, 2014). As
       a response, six siphons were installed in 2011 to temporarily reduce lake volume prior to the implementation of a permanent
       engineering solution. This project progressed slowly in the context of institutional instability, and volume regulation remains
       ongoing in 2019 with a set of 10 siphons.

       With a volume of $17.403 \cdot 10^6$ m$^3$ in 2016 (i.e., +3,380% in 42 years; UGRH, 2016), Lake Palcacocha is among the largest

moraine-dammed lakes in the Cordillera Blanca. Further potential for lake growth is, nevertheless, limited by topographic
       constraints of lateral side moraines and bedrock slope reached in the rear part of the lake. According to the recent lake inventory
       and GLOF susceptibility assessment (Emmer et al., 2016), Palcacocha is among the lakes susceptible to produce a GLOF,
       which could be triggered by rapid landslide processes from surrounding moraines, ice and rock slopes. Detailed study of
       potential landslide-induced outburst flood was performed by (Klimeš et al., 2016) and flood and inundation hazard modelling

for Huaraz was published by (Somos-Valenzuela et al., 2016) and (Frey et al., 2018), suggesting a decreasing hazard level if
       the water level is lowered. The population of Huaraz has increased from approximately 12,000 residents at the time of the
       1941 GLOF to about 140,000 today, with tens of thousands inhabiting the path that the 1941 followed along the Quillcay
       River.

**3 Physical drivers of risk**

In this section we explore to what extent flood hazard and risk from lake Palcacocha can be attributed to anthropogenic climate
       change and to other physical drivers of risk. This is a challenging task with hardly any precedence and first needs some
       conceptual considerations, drawing on recent understandings of how impacts can be attributed to (anthropogenic) climate
       change (Cramer et al., 2014; Stone et al., 2013). A formal attribution study investigates whether a particular system has shown
       any observable trend and whether this trend can be attributed to anthropogenic climate change. Figure 4 visualizes a cascade

of impacts from anthropogenic emissions to climate change, glacier shrinkage and lake growth, and eventually to GLOFs and
       resulting flood hazard and damage. If we want to decipher the influence of climate change on GLOF hazard we need to analyze
       each component of this cascade of impacts, considering that a varying number of confounding factors (i.e. factors not related
       to climate) interact at each stage.

       In this cascade, we start with climate change where attribution research has a long and advanced track record and would

typically conclude with a statement such as to what degree the observed climatic changes or trends can be attributed to
       anthropogenic emissions (Bindoff et al., 2013; Stott et al., 2000).





Specific studies on the attribution of observed climatic trends in the tropical Andes to anthropogenic emissions hardly exist so far, and much less for the Cordillera Blanca or the Palcacocha area. Global-scale attribution studies and assessments, however,
have considered the broader Andes and Pacific coastal region. (Bindoff et al, 2013; Jones et al., 2013) show that temperature changes in this region are broadly in line with climate model runs including anthropogenic forcing and clearly deviate from model runs with natural forcing only. Further research has analyzed the observational temperature and precipitation record of the region over the past decades and the link to phenomena of climatic variability such as the El Niño Southern Oscillation (ENSO) (Heidinger et al., 2018; Vuille et al., 2008). (Schauwecker et al., 2014; Vuille et al., 2015) concur in that temperatures
in the Andes of Peru, including the Cordillera Blanca, have increased since the beginning of the observational record in the 1960s at rates of about 0.2 to 0.3°C per decade, with reduced warming rates during the last ca. 30 years (~0.1°C per decade). While ENSO and the Pacific Decadal Oscillation (PDO) exert an important influence on an interannual or decadal scale, anthropogenic radiative forcing has been identified as the most likely cause of the longer term warming (Vuille et al., 2015).
We proceed along the impact cascade (Fig. 4) with glaciers. Glaciers are closely coupled to the climate system, but surprisingly,
there exist only very few studies worldwide that explicitly attribute glacier change to anthropogenic climate change. If we revisit glacier decay in the Cordillera Blanca, including the Palcacocha area, we find a phase of rather strong glacier retreat in the late 19th century, followed by a slow-down in the early 20th century with small advances in the 1920s (Kinzl, 1969). Later, a period of strong glacier shrinkage in the 1930s and 1940s led to a phase of slow retreat in the 1950s to 1970s, eventually followed by very marked glacier loss since the late 1970s until present (Georges, 2004; Hastenrath and Ames, 1995; Kaser
and Georges, 1997; Rabatel et al., 2013). The continuous mass loss since the late 1970s was enhanced (or reduced) by variations of the Pacific sea surface temperatures, and El Niño and La Niña phases, respectively, with ENSO exerting a significant effect on Andean glaciers on interannual time scales. The long-term glacier shrinking trend, however, cannot be explained by ENSO-related variability (Schauwecker et al., 2014; Vuille et al., 2015), and therefore climate change clearly plays a significant role. This is also reinforced by the IPCC who attributed glacier retreat in the Andes to climate change with
very high confidence (Magrin et al., 2014).
The only available quantitative glacier attribution study that also includes the tropical Andes concludes that globally more than two thirds of the 1991-2010 glacier mass loss is due to anthropogenic forcing, and for tropical regions finds that an anthropogenic signal in observed glacier mass loss of recent decades is detectable with high confidence (Marzeion et al., 2014). The anthropogenic signal is much stronger for the past 2-3 decades as compared to earlier time periods.


We now analyze how Palcacocha lake growth relates to glacier shrinkage and anthropogenic climate change. Lake Palcacocha extends on a relatively flat area that was previously occupied by glacier ice, and is dammed by LIA and early 20th century moraines. Lake growth at Palcacocha can therefore be attributed to glacier retreat in a straightforward way as glacier ice was simply replaced by lake water, and close to 100% of the lake growth can be explained by glacier retreat (Fig. 2). Thermal
energy of lake water accelerates ice mass loss at the glacier front, generating a positive feedback between glacier retreat and lake growth (Kääb and Haeberli, 2001). Lake growth was highest in the 1990s and 2000s (Fig. 3), coinciding with the period where glacier shrinkage can regionally be attributed to anthropogenic emissions with high confidence. We therefore conclude that growth of lake Palcacocha has a clear anthropogenic signal.



How GLOF hazard in Huaraz or elsewhere can be attributed to anthropogenic climate change has conceptually and scientifically not been clarified so far. Physically, flood risk in Huaraz is determined by GLOF hazard which is a function of the magnitude (or intensity, such as flood height) of a hazardous process at a given location, and its probability of occurrence (Raetzo et al., 2002; UNISDR, 2009). A number of factors influence and determine GLOF magnitude and probability of occurrence at lake Palcacocha, notably lake volume, dam stability and freeboard, and landslides from unstable moraines or

ice/rock avalanches impacting the lake (Emmer and Vilímek, 2013; Schneider et al., 2014). Some of the factors (such as lake formation) are closely related to climate change while others can be associated to geologic or geotechnical conditions (e.g. dam stability), or are explicitly influenced by human intervention aiming at reducing the risk of GLOFs (e.g. lake freeboard determined by the height of the constructed drainage canal). In addition to effects on glacier retreat, climate change can influence some of these hazard-determining factors, e.g. increasing temperatures can degrade permafrost and thus destabilize

the flanks of the steep headwalls surrounding lake Palcacocha, or alter thermal conditions and stability of steep glaciers (Carey et al., 2012; Faillettaz et al., 2015; Haeberli et al., 2017).

To assess how GLOF hazard at lake Palcacocha translates into flood hazard in the city of Huaraz we draw on numerical mass flow simulations by (Frey et al., 2018) and (Somos-Valenzuela et al., 2016), who modeled different scenarios of avalanches impacting the lake and producing dam overtopping waves and downstream propagating floods (see Suppl. Material). They

follow state-of-the-art hazard assessment approaches (GAPHAZ, 2017), recently also applied to others lakes in the Cordillera Blanca (Schneider et al., 2014). Corresponding model results indicate that an urban area of similar size as destroyed by the 1941 GLOF is threatened by high GLOF hazard and thus by potential devastating effects (Fig. 5). Previous studies estimated about 40,000 people living in the inundation zone with a potential death toll of close to 20,000 (Somos-Valenzuela, 2014).

At the current state of science, an assessment of GLOF hazard attribution to anthropogenic climate change can only be

qualitative. Although, as seen above, non-climatic factors also influence GLOF hazards, we can confidently state that in the absence of anthropogenic climate change the flood hazard would be much lower, primarily because the size of the lake would be substantially smaller and a longer, flat glacier tongue, as it was the case in 1941, significantly attenuates the impact energy of potential ice or rock/ice avalanches (Mergili et al., 2020).

## 4 Socio-economic drivers of risks

While physical drivers of GLOF hazard such as climate change, ice loss, and glacier lake expansion increased risk in the valley below Lake Palcacocha, many societal drivers of risk have simultaneously intersected with geophysical changes and have exacerbated vulnerability and people's exposure in Huaraz. Socio-economic status, governance and institutional aspects, technology and knowledge production, and cultural forces have all influenced GLOF risk from Palcacocha. For one, risks stem from the placement of the city of Huaraz and its ever-increasing population at the confluence of the lower Quillcay River and

the Santa River, where several Cordillera Blanca lake basins drain. Spanish colonists initially founded Huaraz in the sixteenth century, preferring to build their towns on valley floors in riparian zones, a pattern that contrasted with pre-Columbian populations that implemented a form of hazard adaptation by settling in upland areas away from alluvial fans (Oliver-Smith,





1999). The 1941 Palcacocha GLOF illustrated the consequences of this placement and the city's long-term exposure to Cordillera Blanca hazards (Wegner, 2014). Following the flood, authorities attempted to reduce hazard-zone inhabitation by
prohibiting construction in the GLOF path, but residents and newcomers ignored the hazard zoning policies and the government did not enforce its mandate (Carey, 2010). After the devastating 1970 earthquake destroyed much of Huaraz, the government again prohibited reconstruction in the 1941 GLOF path due to new concerns about unstable glacier lakes above Huaraz (Bode, 1990; Carey, 2010; Oliver-Smith, 1986). Once again, residents defied government hazard zoning, both rebuilding downtown Huaraz and expanding upstream along the banks of the Quillcay River toward Palcacocha and other
glacier lakes. According to flood hazard assessment and mapping presented in Figure 5, the Huaraz inhabitants most exposed to a future Palcacocha GLOF cluster along the Quillcay river in the districts of Nueva Florida, Antonio Raimondi, Centenario, parts of San Francisco, Huarupampa, Nicrupampa, José Olaya and Patay, which have largely expanded in the past decades (Bode, 1990; Carey, 2010; Wegner, 2014). Figure 6 spatially compares the urban area of Huaraz from the immediate aftermath of the 1941 GLOF to the current situation, revealing enormous urban growth including the most hazard-exposed areas. Census
data from a similar timeframe also shows an enormous population increase from about 11,000 in 1940 to more than 140,000 in 2017 (Fig. 7). Several reasons motivated inhabitants to resettle and build within the potential path of a Palcacocha GLOF, even though they recognized the GLOF risks. Analysis of these reasons helps illuminate socio-economic drivers of GLOF risk that are useful not only for understanding Palcacocha, but also for evaluating GLOF and hazard risk worldwide.

First, inhabitants recognized key economic factors: some believed they would incur direct economic losses if they moved away, while others thought that inhabiting the area along the Quillcay River adjacent to Huaraz would yield economic gains. This dynamic emerged as early as the 1940s, and residents were outspoken about defending their rights to live in the potential GLOF path—often based on economic reasoning—starting in the 1950s (e.g. Anonymous, 1956, 1951, 1945). Inhabitation of flood-prone areas and other places susceptible to natural disasters, even when people understand the risks, is not unusual (e.g.
Steinberg, 2000; Wisner et al., 2004). In Huaraz, however, many worried that the government would not compensate them for their lost land or provide them with a comparable plot and home elsewhere. Others were concerned that relocation of the city or even moving upslope to safer terrain would diminish Huaraz's position as the region's financial hub, where jobs and markets offered opportunities, transportation and commercial centers attracted people, and banks and credit institutions existed (Doughty, 1999; Oliver-Smith, 1977, 1999). While many were reluctant to leave Huaraz for these economic reasons, others
migrated into the city for related motives, such as receiving relief and disaster aid following the catastrophe (Walton, 1974; Wrathall et al., 2014).

One part of Huaraz, the Nueva Florida district adjacent to the Quillcay River, exemplifies these economic incentives outweighing GLOF risks. Ethnographic research we conducted in the area provided insights that exemplify the historical and
contemporary factors playing into this dynamic. Quechua-speaking farmers from the highlands above Huaraz began buying inexpensive property in Nueva Florida after the 1970s. This previously vacant land was not only affordable but also offered proximity to employment, public services, and overall a higher standard of living for historically marginalized people. In the 1990s, new multinational mining operations near Huaraz triggered an influx of mine workers who frequently settled in Nueva



Florida. Given the district's growth, authorities built paved roads and installed electricity and sewage networks in Nueva Florida in the early 2000s. Today, Nueva Florida is a flourishing district, attracting even more people to the area along the Quillcay River. While authorities have officially prohibited construction in Nueva Florida since Palcacocha GLOF risk concerns arose again in 2009, residents attest that officials tolerate the construction of smaller buildings. Over time, living in Huaraz provided a unique opportunity for Quechua-speaking villagers to access social and economic opportunities in Huaraz. According to a survey we conducted in 2017 (see Suppl. Material), most Nueva Florida residents showed little concern for the

risk of flooding, neither in the past nor today. Though many were aware of recent public and media discussions about the threat of a Palcacocha GLOF, they contended that such warnings were exaggerated. It appears that economic and material benefits of inhabiting Nueva Florida outweigh the potential flood risk.

Second, social status among Huaraz residents—influenced primarily by racial and class divisions—has been another key factor

influencing GLOF risk and explaining some inhabitants' continued occupation of the Quillcay riparian zone. Cities like Huaraz have long been inhabited by the ruling classes – the Spanish-speaking residents and supposedly non-indigenous people (Oliver-Smith, 1999). Living higher and more rural, on the other hand, signified a poorer, more indigenous status in this culturally-constructed schematic of race-class dynamics (Walton, 1974). Post-disaster urban zoning after the 1941 GLOF and 1970 earthquake that attempted to relocate populations to safer ground higher above the river came to symbolize, for some, a

government-imposed assault on ruling class privilege, downward social mobility and loss of socioeconomic status (Bode, 1977, 1990; Carey, 2010; Doughty, 1999).

Analysis of GLOF risks, exposure and vulnerability must consider both how inhabitants rank their risks and how disaster prevention policies such as hazard zoning, building practices, and urban planning affect socio-economic status. It is difficult

to pinpoint responsibility for people's decisions to inhabit the potential GLOF path below Palcacocha. Inequality driven by class and race divisions has led to the marginalization of some segments of the Peruvian population. As a result, their decision-making may be shaped by economics, livelihood and employment opportunities, social standing, and other socio-economic factors that are usually impossible to attribute to certain individuals but rather to larger forces such as racism, poverty, and global inequality. Global histories of colonialism, neoliberalism, resource extraction, political domination, and economic

marginalization also make Peruvians poorer compared to residents of the most developed nations in the Global North, who tend to have lower levels of vulnerability and can afford to rank risks differently than Peruvians living beneath lake Palcacocha (Carey, 2010).

## 5 Institutional and governance-related risk drivers

Institutions, policies, and governance also affect levels of GLOF risk. In particular, government instability, fluctuating support

(funding and resources), and institutional inconsistency creating confusion about disaster-prevention roles and responsibilities have all exacerbated risk below lake Palcacocha. It initially took ten years after the 1941 Huaraz disaster to form the first GLOF-prevention office, the Control Commission of Cordillera Blanca Lakes (CCLCB). Since establishment of the first



glaciology and lake security office in 1951 to mitigate Cordillera Blanca GLOF risks, the agency has passed through four
different ministries, had twelve different names, and even disappeared completely for nearly four years in the late 1990s (Carey,
2010). Some disaster events (e.g. 1950 Los Cedros GLOF, 1970 earthquake and Mount Huascarán avalanche) and some
authoritarian governments (e.g. Presidents Odría in the 1950s and Velasco in the 1970s) stimulated strong investments in
Cordillera Blanca GLOF prevention. At other times, glacier disasters (1962 Ranrahirca avalanche) and authoritarian
governments (President Fujimori in the 1990s) triggered little government response or even backward steps in GLOF risk
reduction.


Decentralization of the national government has also exacerbated institutional inconsistency and instability, which also
influences GLOF risk. Prior to the 2002 start of the decentralization process, Peru's 25 departmental governments functioned
as administrative extensions of the national government, with departmental governors (prefects) appointed by the national
government. During this period, the central government directed and consolidated Cordillera Blanca GLOF monitoring and
mitigation. Decentralization created new, more autonomous regional governments that were elected (Arce, 2008; Dickovick,
2011). On paper, the reforms made the Ancash Regional Government primarily responsible for identifying and implementing
Palcacocha risk reduction measures. But in practice, decentralization generated confusion about jurisdiction, expertise,
authority, funding, and responsibility, often leading to stagnation and non-action that left residents more vulnerable or exposed
to potential GLOFs. Amidst decentralization, the Ancash government has also experienced exceptional turmoil in recent years:
since 2014, three governors of Ancash have been imprisoned over charges including corruption and assassination (El Comercio,
2018). Further, there remains a host of national government institutions and ministries with jurisdiction over the Cordillera
Blanca, including the Glacier and Lake Evaluation Office (formerly Glaciology and Water Resources Unit, UGRH) of the
National Water Authority (ANA) and associated local and provincial water authorities, and Huascarán National Park. They
interact with the Ancash Regional Government, provincial and municipal authorities and their corresponding entities such as
civil defense, rural community jurisdictions (*comunidades campesinas*), and a host of other stakeholders including mining
companies, Duke Energy, and non-governmental organizations (NGOs). More specifically for GLOF risk reduction, the
national government agencies ANA, and the National Institute for Glacier and Mountain Ecosystem Research (INAIGEM),
founded in 2015, operate in the Cordillera Blanca but sometimes overlap in confusing ways, ultimately impeding institutional
capacity to respond to increasing glacier risks.


This regional government instability and uneven decentralization has obstructed effective GLOF risk reduction measures at
lake Palcacocha specifically. In 2003, Palcacocha overflowed and caused a small flood due to a landslide into the lake (Vilímek
et al., 2005). While debate ensued about jurisdiction and responsibility (e.g. Congreso de la República, 2003), it took nearly a
year to conduct a bathymetry study and repair the damaged flood protection dam at the lake. In 2009, when a new study
revealed that Palcacocha contained 17 million m$^3$ of water (more than it had for the 1941 GLOF), no single institution took
charge to lead a permanent engineering project to partially drain and secure the lake, as the UGRH had done for decades in the
past. Instead, the institutional instability generated only short-term, unsustainable measures (temporary siphons) to protect
downstream populations, despite repeated studies documenting Palcacocha risks (Hegglin and Huggel, 2008; Klimeš et al.,





2016; Portocarrero, 2014; Somos-Valenzuela et al., 2016; Vilimek et al., 2005). In response to political inaction at a regional
level, the local governments of Huaraz and Independencia – the two main municipalities affected by GLOF risk from
Palcacocha – have collaborated to implement a Palcacocha early warning system. Moreover, in 2016, international experts in
cooperation with local institutions released a new hazard map, including GLOF hazards and evacuation plans, for the Quillcay
catchment (Frey et al., 2018), cf. Section 3 above). International scientific institutions and NGOs took primary charge of
producing the map, in collaboration with, but without leadership of local, regional, or national institutions in Peru. Overall,
combined effects have contributed to the increase of risk, namely related to decentralization of the national government,
institutional instability, conflicting roles and jurisdictions, and waning government support for Palcacocha hazard reduction
research, monitoring, and projects. Given the complexities surrounding these processes and dynamics over time a more detailed
indication of their contribution to risk is elusive.

## 6 Cultural and emotional components of risk

Cultural factors also influence risk in the valleys below lake Palcacocha. Attachment to place can motivate people to inhabit
potential flood zones, while varying local explanations of cause-effect (particularly causation between human behavior and
environmental change) can also yield certain understandings of risk that collide with scientific assessments and may lead to
inaction in the face of GLOF risks. Research on these cultural dimensions of glaciers is growing, elsewhere (Allison, 2015;
Cruikshank, 2005; Sherpa, 2014; Sherry et al., 2018), and in the Peruvian Andes and Cordillera Blanca, where locals often
perceive sentient landscapes and maintain spiritual relationships with mountains and glaciers (Bolin, 2009; Carey, 2010; De la
Cadena, 2015; Jurt et al., 2015). One key cultural driver of risk along the Quillcay River is the emotional and psychological
attachment to place that has historically attracted people to Huaraz, even after the 1941 GLOF and 1970 earthquake devastated
the city. A profound sense of place – that is, attachment to homelands, personal identity, heritage, familiarity with landmarks
and landscapes, and links to community – frequently bonds people to particular places, not just in areas prone to GLOFs but
in disaster zones worldwide (Hastrup, 2013; Oliver-Smith, 1982; Sherry et al., 2018). These attachments to land and community
also motivated people to remain living in Huaraz, even after disasters struck or when they had knowledge of GLOF risks
(Bode, 1990; Oliver-Smith, 1982, 1986; Yauri Montero, 1972). While some survivors emigrated to Lima after the 1941 and
1970 disasters, others remained in their former homeland, connected to their birthplace, close to those who died in the disasters,
and part of the same community where they had always lived and experienced trauma.


Another factor influencing risk is the diverse understandings of environmental processes and hazards, particularly where
scientific and technical explanations contrast with local beliefs and values. In May 2017, two ice avalanches descended into
lake Palcacocha within a 24-hour period, causing three-meter high waves that lake workers witnessed. The workers' supervisor
maintained that this event occurred because he had not paid tribute to Palcacocha and the surrounding mountains. For workers
at Palcacocha and other Quechua-speaking farmers living nearby, the lake and mountains are beings that require respectful
engagement. According to this understanding of glaciers and lakes, spiritual disruptions could trigger a GLOF – such as lake
workers' inadequate offerings to mountain beings, rather than only geophysical processes such as glacier and bedrock





instability. Some local accounts voiced that past glacier-related disasters such as the 1941 GLOF occurred because people failed to show the landscape entities adequate respect (Yauri Montero, 2000). According to our interviews and focus groups

we conducted in 2017 and 2018, some elderly villagers in areas below Palcacocha corroborate these stories. In one of these local's accounts of the 1941 flood,[2] a deity told a rural woman to perform a ritual offering at Palcacocha. When she failed to do so, the lake became angry and flooded Huaraz. Asked why Cordillera Blanca glaciers are melting, lake workers at Palcacocha pointed to contamination and global industry. While they recognized a global dimension of environmental change, they regularly paid tribute to the lake and mountains in an effort to prevent disaster. As long as the supervisor kept the lake

happy with offerings of coca leaves and alcohol, he explained, there would be no GLOF disaster.

These accounts thus reveal how local people perceive both global and local aspects as drivers of risk, but their perceptions are often not in line with technical and scientific assessments of risk. For instance, many urban and rural residents have referred to enchanted lakes, which, in local understandings, can lure people to their shores and then suck people inside, to the other

world, if they do not perform proper rituals or resist approaching these lakes (Carey, 2010; Yauri Montero, 2000). Other residents offer different cultural explanations for natural disasters, such as Catholic residents saying that the 1970 earthquake resulted from sinners' behavior and God's will (Bode, 1990; Oliver-Smith, 1986). Attributing GLOFs to their neighbors' behaviors or to the will of certain deities can ultimately lead to a relinquishment of responsibility and fatalism: why move outside a potential GLOF path if floods are determined by God's will or neighbors' sins? When a resident believes sinning

causes floods or coca leaf offerings presented to mountain deities stabilize glacier lakes – as opposed to the scientific conclusions attributing these processes to climate change, glacier shrinkage, and bedrock geometry – then development and implementation of risk reduction plans become more difficult, because not everyone agrees about the source of the hazard. In fact, people in Huaraz negotiate cultural and scientific understandings of flood risk on a daily basis, and may regard multiple explanations as valid.

**7 Implications for responsibility and justice**

So far we have examined physical climate change related, socio-economic, institutional and cultural aspects of Palcacocha GLOF risk. Drawing on that, we now analyze the possible implications for responsibility, and ask how concepts of justice can inform these and other similar issues.

Responsibility as a concept commonly concerns four aspects that become relevant when analyzing the differentiation and

assignment of responsibilities in specific circumstances and at different policy levels (Bayertz, 1995): i) Someone (the agent or subject of responsibility) is responsible for ii) something (the object of responsibility) and answerable to some iii) institution according to some iv) norm.

Responsibility often concerns different agents and objects (Wallimann-Helmer, 2016). In the case of Palcacocha, a complex network of responsibilities and dependencies exists between different agents of responsibility and institutions. Differentiation

and assignment of responsibilities to subjects depends on the perspective of the different drivers of GLOF risk, and on whether

---

[2] Interview conducted in 2017



a forward- or backward-looking concept of responsibility is adopted (Miller 2007). Backward-looking assignment of responsibilities identifies the agents bearing responsibility for risks and outcomes already materializing. Forward-looking ascription of responsibilities concerns remedial duties to prevent negative impacts or minimizing risks (Burns and Osofsky, 2009; Grossman, 2003).

Observed physical risk drivers indicate that to a large extent glacier shrinkage and lake growth are due to anthropogenic climate change that contribute to GLOF risks. Detection and attribution research is primarily a backward-looking science and may inform the assignment of responsibilities for past emissions causing present climate risks (Huggel et al., 2016; James et al., 2019). Historically, emitters contributing to climate change are primarily countries and regions from the industrialized global north. Accordingly, detected and attributed physical risk drivers of GLOFs allow us to ascribe some responsibilities for
increased risk of GLOFs to these countries and regions.

Attribution research has only limited explanatory value for assigning forward-looking responsibilities, which also depends on the extent to which specific future risks are controlled by past emissions and related environmental changes. For instance, lake Palcacocha has formed as a result of climate and glacier change of the past decades but is likely to persist for decades or even centuries into the future. Assignment of forward-looking responsibilities in case of climate-related loss and damage commonly
implies remedying negative impacts or minimizing the risk of their occurrence, i.e. in case of Huaraz minimizing risks of GLOFs and their impacts (Wallimann-Helmer et al., 2019). Investigating the different risk drivers can be useful to identify what risk reduction measures need to be taken but it cannot identify the appropriate responsibility bearers, nor whether remedial responsibilities should concern monetary payments, help in building the required infrastructure and protection measures, assistance in governance or capacity building.


It seems plausible that industrialized countries and regions contributing most to anthropogenic climate change foster the development of appropriate infrastructure and capacity in order for the affected people to be able to govern local climate risk themselves. In cases like Huaraz, this is particularly important for two reasons. Firstly, many locals moved to Huaraz and especially to Nueva Florida for social and economic reasons. As we have seen, relocation out of the GLOF hazard zone means
to many a risk of losing social status and achieved assets, exacerbated by a lack of trust in the government to compensate people so that they can retain their achieved status. Capacity building here demands building trust in governance institutions and, if necessary, providing financial resources. Secondly, due to the socioeconomic opportunities provided by moving to Huaraz from rural areas as well as due to cultural beliefs, perceptions of GLOF risk are diverse and not necessarily congruent with technical and scientific findings. This makes the sharing and exchange of comprehensive information and education to
inhabitants of Huaraz and especially to those living in the flood hazard zone another key factor of capacity building. Otherwise, there is a risk of decision-making by locals on the basis of insufficient information. Local or international experts may provide information on what can happen in case of a GLOF (e.g. flood height and extent in Huaraz). However, for reasons of efficiency and effectiveness, and of local appropriation and acceptance of measures, it is sensible to leave decisions about what constitutes an acceptable or tolerable risk and how risk governance is implemented to those people who are most directly affected (Kaswan,
2016). In fact, locals' perspectives (e.g. in terms of cultural and spiritual understandings) should be taken seriously, suggesting





a dialogue of knowledges about GLOF risk and environmental change more broadly, rather than a hierarchical knowledge exchange.

Governance institutions and legal regulations define whether or not and to what extent individuals must bear responsibility for their own decisions with regard to settlements in risk zones like Nueva Florida. Institutions regulate behavior and demand justification if their regulations are not followed. However, institutions themselves are most often also responsibility bearers. The policy level at issue thereby defines the agent to take on responsibilities and the object of the responsibilities to be taken on (Wallimann-Helmer et al., 2019). Institutions are answerable to other, higher-level institutions and depend in their functioning on these institutions. For instance, the Glacier and Lake Evaluation Unit in Huaraz depends on finance and decisions from the central government in Lima through the National Water Authority (ANA), and according to available resources, this office can take on more or less ambitious responsibilities. Since responsible agents are always answerable towards some institution, the institutional inconsistencies and instabilities at Palcacocha tend to foster lax implementation of regulations on the side of the agents that should take on responsibility. Who bears the responsibilities to help establish, strengthen and maintain functioning institutions depends on the governance level and capacities of relevant responsibility bearers. Socio-economically disadvantaged locals might not be able to strengthen institutions but the wealthy ones and government officials may have this power. In case of Palcacocha, some technical and governmental institutions in conjunction with international assistance may be best suited to do so, including the Ancash Regional Government and possibly the municipal government of Huaraz.

These considerations in relation with the Palcacocha case suggest that there may be at least two different perspectives of responsibility, which, however, we consider neither as competing nor mutually exclusive. One of them aligns with the ability to pay principle (Caney, 2005; Page, 2008) that proposes that capacity is the most important criterion for fairly differentiating responsibilities in the context of climate risk governance. One may argue that efficiency and effectiveness in risk governance is achieved if those agents and institutions with appropriate capacity take on responsibility. This perspective would then also call for capacity building efforts where capacity is lacking.

The other responsibility perspective is more guided by the polluter pays principle (Gardiner, 2004; Hayward, 2012), implying that other responsibility bearers would have to carry heavier burdens. In global climate policy, it is the industrialized countries from the global North that have heavily contributed to anthropogenic emissions and are thus assigned heavier burdens. Applying the logic and mechanisms of global climate policy to the Palcacocha case would foresee global emitters nourishing international climate funds (such as the Green Climate Fund), used to implement local adaptation and risk management measures. With reference to the ongoing court case over Palcacocha GLOF risk, this implies that RWE would be held responsible because, albeit the company's contribution to Palcacocha GLOF risk is proportionally small and hardly quantifiable, it is undeniable. However, we have also seen that a substantial, yet again hardly quantifiable fraction of increased GLOF risk in Huaraz is due to socio-economic, institutional and cultural factors with a complex network of agents and responsibilities. This suggests that while global emitters bear responsibility for their contribution to locally materializing risks, local governments are not exempted from their responsibilities to address and effectively reduce the risk of negative GLOF impacts.





## 8 Conclusions

Palcacocha is in many aspects representative for the interlinkages of global and local drivers of climate risks and potential or actual loss and damage. The case shows that risks develop, and loss and damage occur, in a local context and over a certain period of time. Comprehensively understanding the different contributors to risk is challenging and has only been addressed
by research in a limited way so far. Risk (and associated loss and damage) is a multi-faceted construction and the question of causality can often not be fully solved, at least not in a quantitative way.

Here we have seen that an anthropogenic signal (related to GHG emissions) is traceable through an impact chain of temperature, glacier change and associated lake growth that has increased GLOF hazard in Huaraz over the past few decades. Long-term climate, glacier and lake observations, modeling, geotechnical and geomorphological analyses and flood modeling
are needed to develop an understanding of the impact cascade. In contrast, the current conditions of exposure and vulnerability of people and values in Huaraz to GLOF hazard can only be understood with a historical perspective of social, economic, political and cultural dynamics.

Questions of responsibility or even liability are difficult to be addressed in such cases where global, national and local drivers build a complex interlinked network. Courts, as in the case of Luciano Lliuya vs RWE, operate under specific rules of (national)
law that we have not further analyzed here. For questions of responsibility, we have sketched how the Palcacocha case could be embedded in a normative framework where we distinguish between perspectives of efficiency (with respect to risk management) and contributor pays principles. Rather than promoting one or the other principle we suggest a more differentiated and blended perspective on responsibility, implying that global emitters commit to support strengthening capacities in affected regions and localities, and local institutions and societies engage in local risk reduction measures and
policies.

While such models are in line both with developing country research (Miller, 2007) and international climate policy, the Palcacocha case shows how deficient these mechanisms and efforts still are. An improved understanding of drivers and explicit differentiation of responsibilities could contribute to more effectively addressing climate risk and loss and damage

## Competing interests

Two authors had paid and unpaid working relation with German Watch, the non-governmental organization that supports Saúl Lliuya in the court case Lliuya vs RWE. Specifically, N.WC. had an internship with German Watch (2 months, 2014), followed by short-termed consultancies (2014 to 2018) and employment in 2016. A.E. prepared scientific material for German Watch on a short-term paid basis, and C.H. made an expert statement for the Lliuya party of the court case on an unpaid basis.

## Author contributions

CH developed and led the study, wrote and edited text, analyzed the climate change attribution part, did the socio-economic data analysis and produced figures. AE contributed the sections and data on the lake development, HF the GLOF modeling and hazard assessments, and both contributed figures. MC and NW-C contributed and wrote the section on socio-economic,





institutional and cultural aspects of risks. IW-H led the section on justice and responsibility. All authors revised and edited the manuscript.

## Acknowledgements

We acknowledge the collaboration and exchange with CARE Peru, the Glacier and Lakes Evaluation Office of the National Water Authority (ANA) and the National Institute for Glacier and Mountain Ecosystem Research (INAIGEM), as well as with the local residents.

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





**Table**

| Period / date | Milestone | Details | Reference |
|---|---|---|---|
| 1590 - 1630 | Glacier advance and formation of the moraine | Formation of the moraine which (later) dammed the lake | (Emmer, 2017) |
| late 1930s | European expeditions to the lake and the first photographs | Hanz Kinzl visited Palcacocha and other Cordillera Blanca glaciers and glacier lakes (1939) | (Byers, 2000; Carey, 2012; Wegner, 2014) |
| 13th December 1941 | Dam failure and GLOF | See main text | (Carey, 2010; Oppenheim, 1946) |
| 1950s - 1960s | Lake stagnation | 1942: clearance of drainage canal to facilitate drainage 1950s: construction of a security dam with a drainage canal at its base to prevent rising lake levels level. Minimal lake growth/ stagnation | Air photographs; Carey 2010 |
| 31st May 1970 | Heavy earthquake | M=7.9 earthquake hit the region, no recorded damages on the dam | (Lliboutry et al., 1977) |
| 1973 - 1974 | Remediation | Lowering lake level by 1 m and stabilizing it at 4,566 m a.s.l.; reinforced and rebuilt the 1950s security dam, including a permanent drainage canal; construction of a second security dam on the terminal moraine | (ELECTROPERÚ, 1974) |
| 1974 | Bathymetry | Volume $0.515 \cdot 10^6$ m$^3$ | ELECTROPERÚ, 1974 |
| 1970s - 2000s | Lake growth | See main text | Zapata et al., 2004 |
| 2003 | Dam overtopping and GLOF | Landslide on left lateral moraine into the lake; partial destruction of secondary security dam, which was rebuilt in 2004 | (Vilimek et al., 2005) |
| 2003 | Bathymetry | Volume $3.690 \cdot 10^6$ m$^3$ | (Zapata et al., 2004) |
| 2003 - present | Accelerated lake growth | See main text | (UGRH, 2016) |
| 2009 | Bathymetry | Volume $17.325 \cdot 10^6$ m$^3$ | (Portocarrero, 2014) |
| 2011 | Remediation | A set of siphons was installed to lower lake level prior to the implementation of permanent solution | Portocarrero, 2014 |
| 2016 | Bathymetry | Volume $17.403 \cdot 10^6$ m$^3$ | UGRH, 2016 |
| August 2018 | Remediation | Lake level lowered by 3 m | Field visits |

**Table 1: Milestones in the evolution of Lake Palcacocha.**


**Figures**

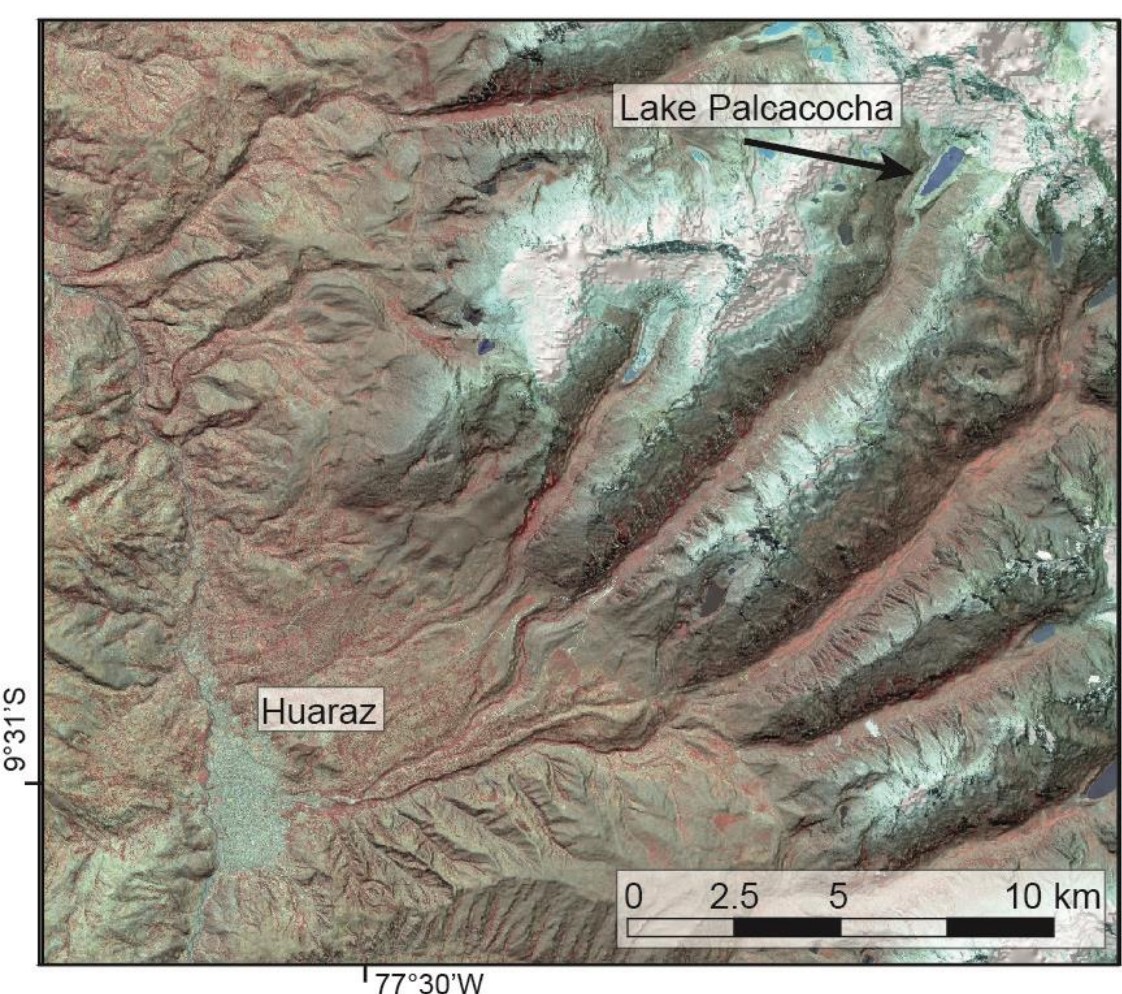

**Figure 1: Study region with Lake Palcacocha and the city of Huaraz (source SPOT image, year of acquisition: 2006).**





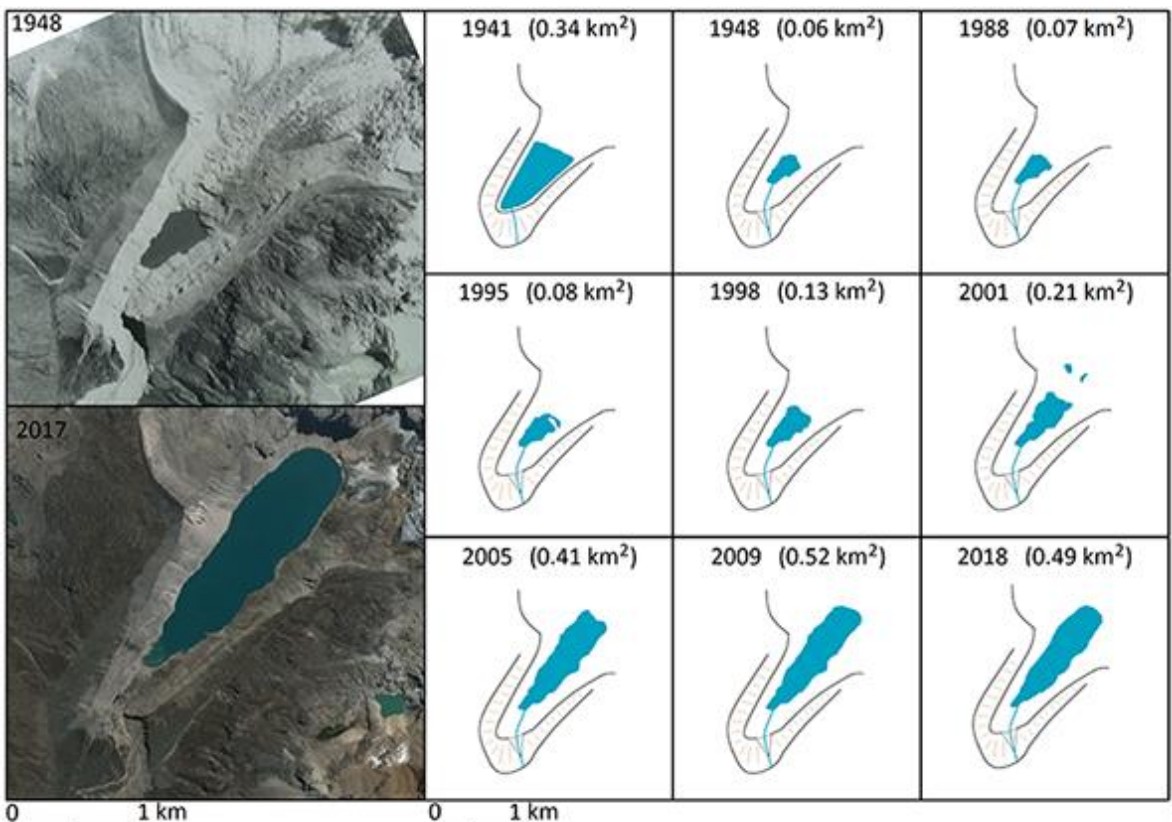

**Figure 2: Evolution of Lake Palcacocha from 1941 to present (source of 1948 image: Archives of the Autoridad Nacional
de Agua, Peru, source of 2017 image: CNES/Airbus image, © Google Earth, date of acquisition: 18 June 2017, source
of lake evolution 1988-2018: Landsat images).**




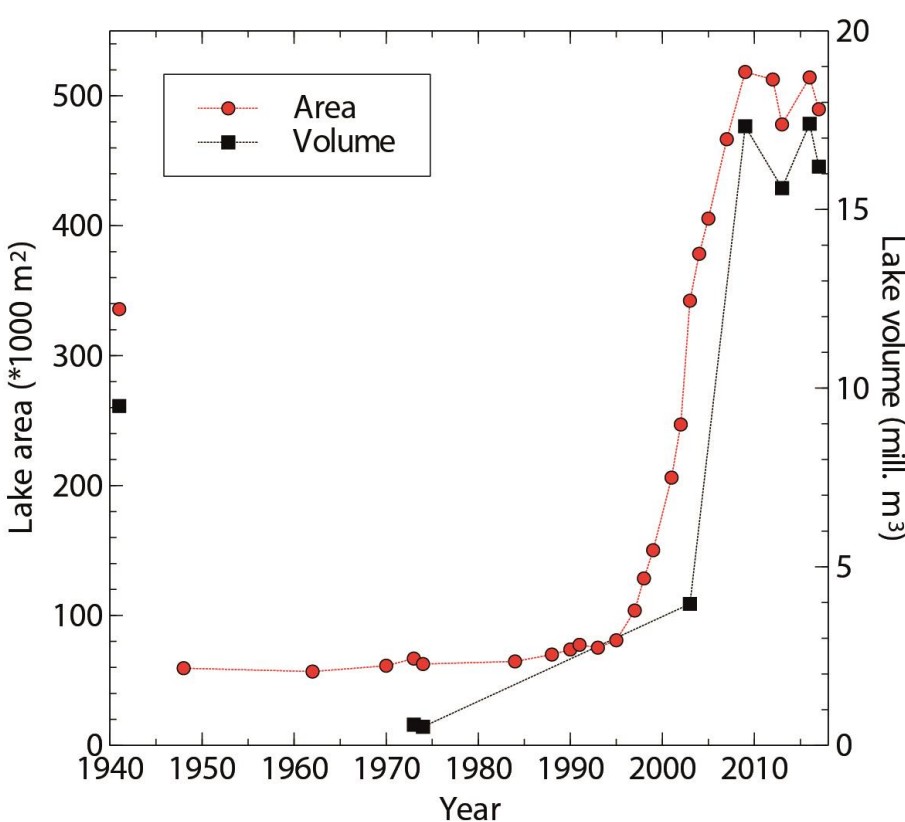

**Figure 3: Evolution of area and volume of lake Palcacocha 1941 to present (data sources: see Table 1).**


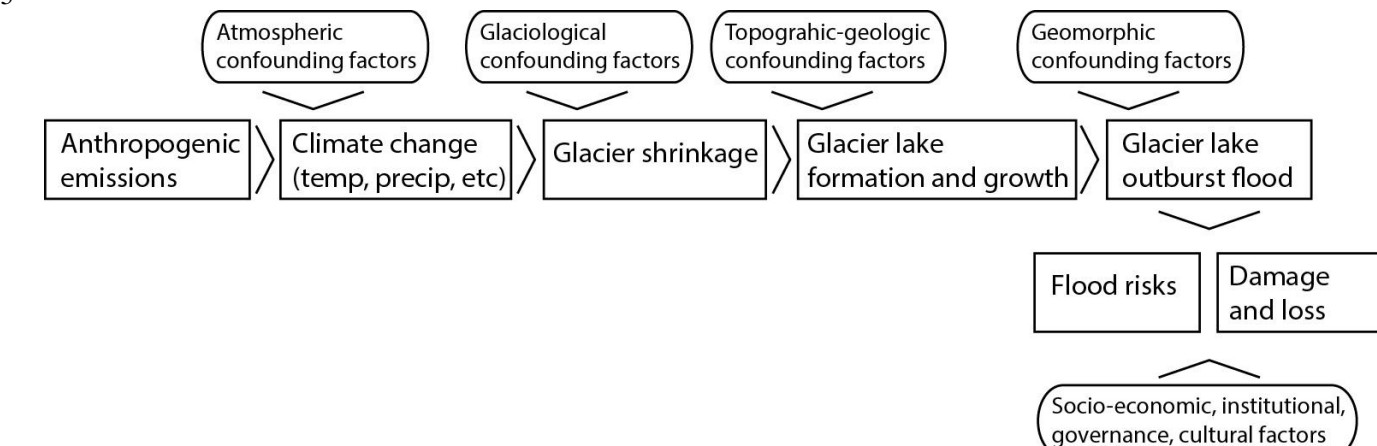

**Figure 4: Causation chain from anthropogenic emissions to glacier lake flood risk. At each element of the causation chain non-climatic (confounding) factors are indicated which also influence the respective element.**




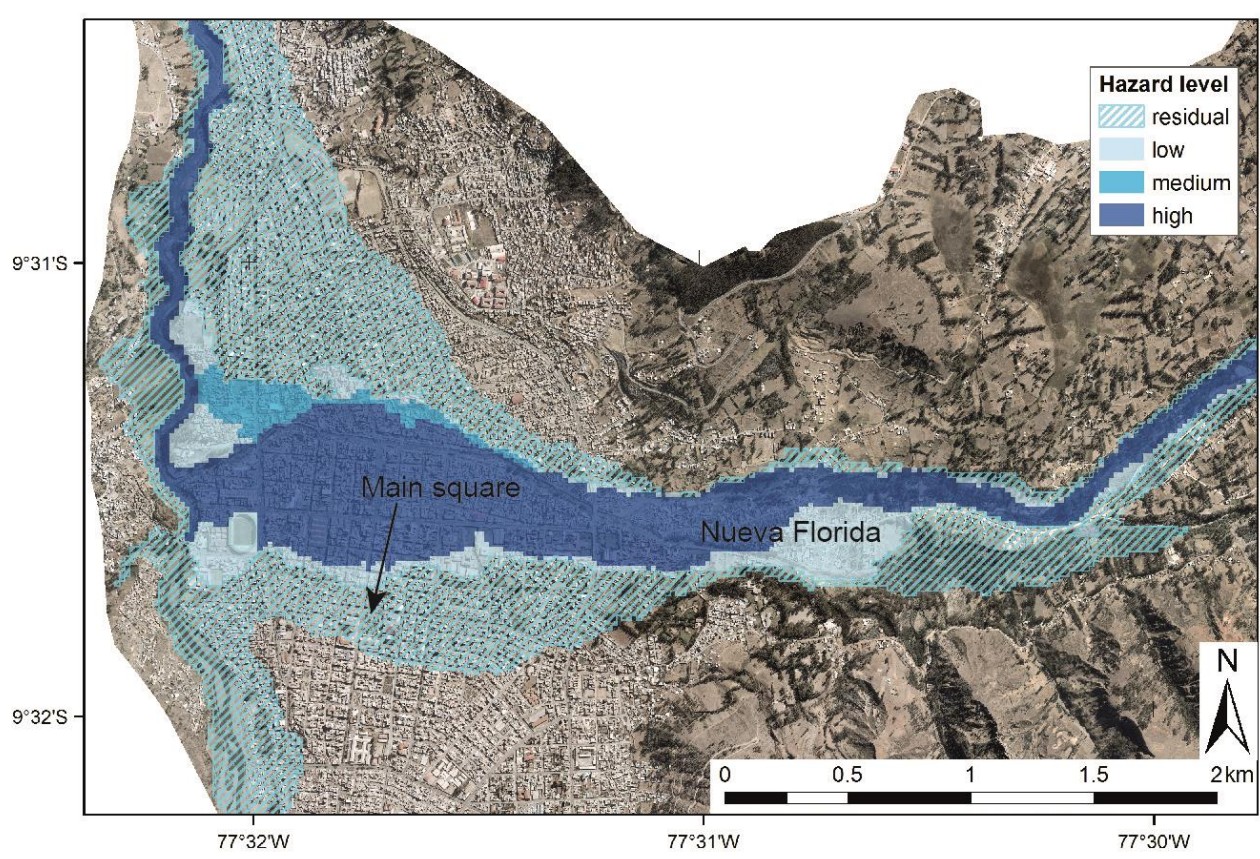

**Figure 5: Hazard map for the city of Huaraz as related to GLOFs from Lake Palcacocha. The district of Nueva Florida and the main square of the city are indicated (for a more detailed description of the hazard mapping methodology see supplementary material) (source of image: © Google Earth / Maxar Technologies, date of acquisition: 11/10/2017).**





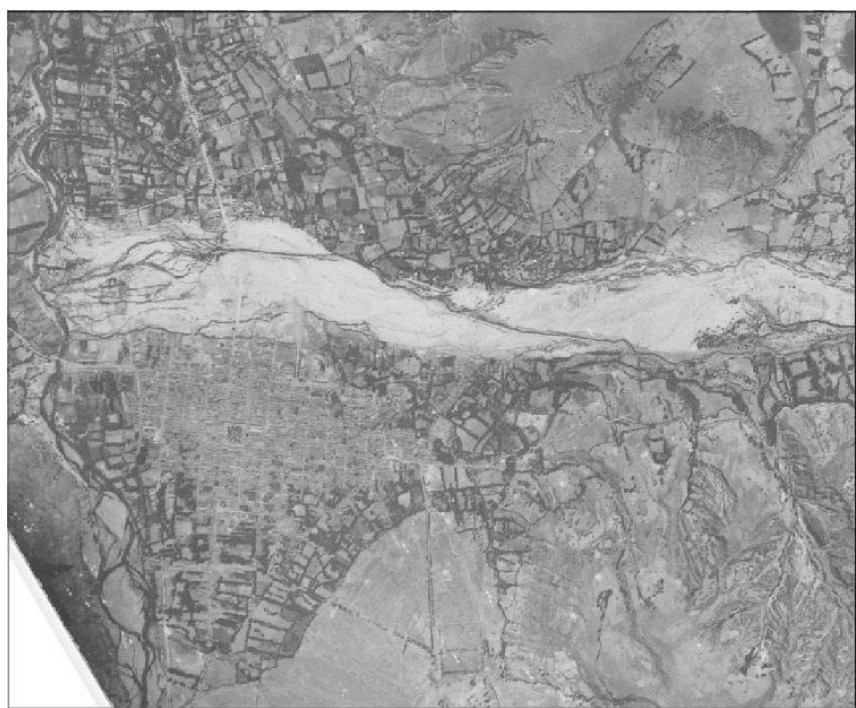

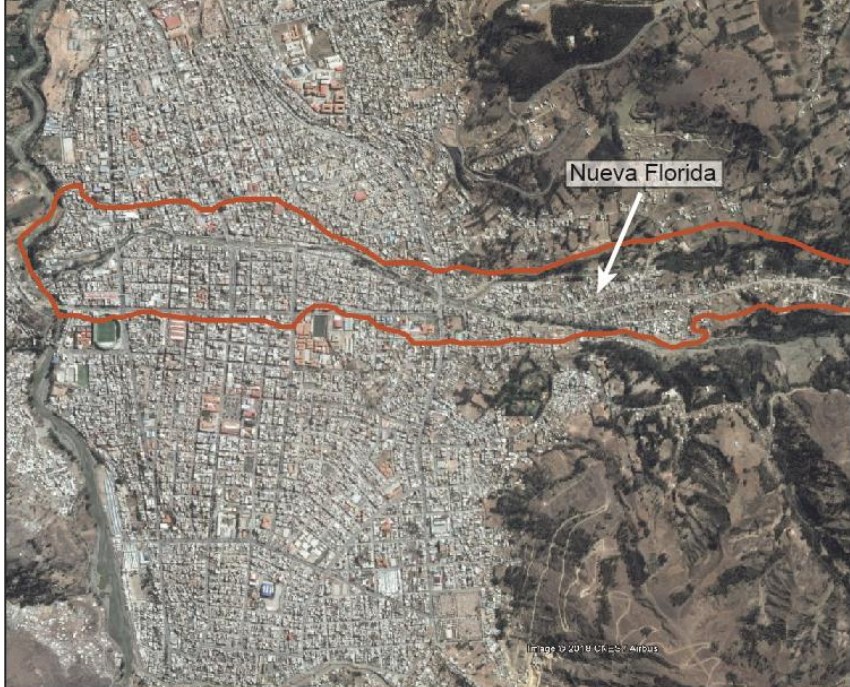

**Figure 6: Aerial photograph from 1948 showing the traces of the 1941 GLOF (upper panel), and 2016 satellite image of the same area (lower panel). The outlines in orange indicate the extent of the area affected by the 1941 GLOF. The highly flood exposed urban district of Nueva Florida is indicated (source of upper image: Archive of Autoridad Nacional de Agua, Peru, year of acquisition: 1948; lower image: © Google Earth / Maxar Technologies, 11/10/2017).**


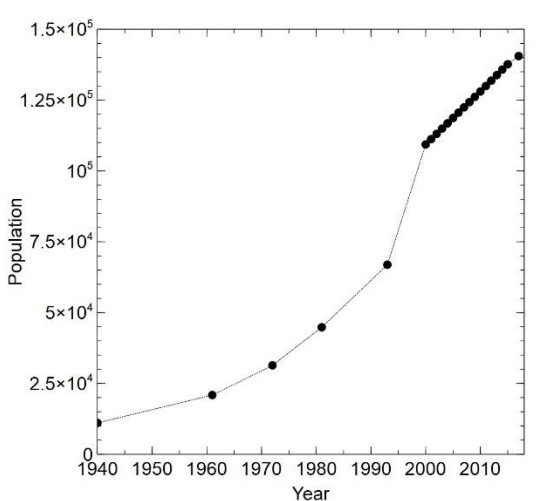
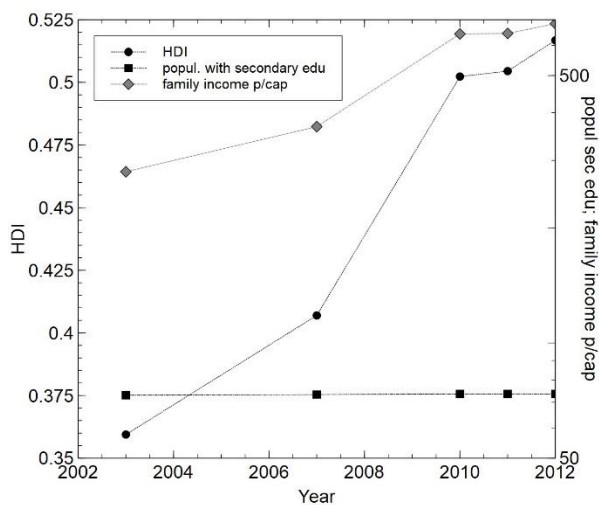

**Figure 7: Population growth of Huaraz over the period 1941-2017. Data since the year 2000 is based on an extrapolation produced by the National Statistical Office of Peru (INEI) (left panel). Vulnerability indicators and their changes between 2002 and 2012 for the city of Huaraz. HDI: human development index, percentage of population with secondary education, and family income per capita in Nuevo Soles per month (data from INEI) (right panel).**

