# Peer review of "Anthropogenic climate change and glacier lake outburst flood risk: local and global drivers and responsibilities for the case of Lake Palcacocha, Peru"

_Natural Hazards and Earth System Sciences, 2020_

## Referee Comment (RC1) · Dmitry Petrakov (Referee) · 30 Mar 2020

General comments: The Lake Palcacocha was a source of the most destructive GLOF in human history which happened in 1941. Some mitigation measures to prevent lake outburst have been done. Growing volume of the lake due to glacier retreat is accompanying by growing population of Huaraz city located at the flow path. Thus any attempts to assess risk of GLOF from the Lake Palcacocha are highly relevant both at local and global scale. Authors used the Lake Palcacocha as a representative case for other glacier lakes and related risks around the world. The paper provides high-quality

case study with significant conclusions, both locally and globally. It also provides nice synthesis of natural and social sciences which is important for comprehensive risk assessments. Novelty of results is quite clear. Structure of the paper is reasonable, Introduction is well-written, motivation and goals of the paper are fully clear. Authors provide brief but comprehensive description of the Lake Palcacocha evolution and engineering solutions to prevent dam failure. Currently lake growth potential is limited by topographic constraints, but outburst probability is high. Authors analyzed anthropogenic contribution to glacier retreat in Cordillera Blanca based on literature review and concluded that growth of lake Palcacocha has a clear anthropogenic signal. GLOF hazard in the absence of anthropogenic climate change the flood hazard would be much lower due to change of lake volume and increasing impact energy of ice/rock avalanches. Socio-economic drivers of risk are determined and analyzed basing on literature review and survey conducted in 2017. What is important, local residents showed little concern for the risk of flooding. The institutional instability generated only short-term, unsustainable measures to protect downstream populations. Authors note that combined effects of institutional and governance-related risk drivers have contributed to the increase of risk and provide interesting insight on cultural and emotional component of risk. Basing on analysis of risk drivers authors provide implications for responsibility and justice.

The manuscript is well written and free of technical errors, well structured, appropriate in length. All figures and the table are high-quality. Supplementary material is very useful for understanding of hazard assessment technology. The conclusions are clear and precise. The results obtained in this study are highly relevant to assess risk of future GLOFs not just in Huaraz but elsewhere. I definitely support publication of the manuscript.

Specific comments:

Authors noted that previous studies estimated about 40,000 people living in the inundation zone with a potential death toll of close to 20,000 (Somos-Valenzuela, 2014).

Anywhere there is no assessment of current situation despite Fig.7 demonstrates significant growth of population in Huaraz. How many people live in zones with different hazard level (Fig.5) now? Is population density within hazardous area uniform or not? An addition of a figure where population density will be provided solely or overlaid to hazard zonation might be interesting for readers and useful for local communities and decision makers. Being fully agree with author's concept I will be happy to see what components of risk are major and what components are really minor. Furthermore, some recommendation how to minimize GLOF risk in Huaraz basing on risk driver analysis will increase practical and intellectual merit of this really great paper.
* * *

---

## Short Comment (SC1) · 6 Apr 2020

Contribution to the Peer Review of

"Anthropogenic climate change and glacier lake outburst flood risk: local and global drivers and responsibilities for the case of Lake Palcacocha, Peru" by Christian Huggel (et al.)

by Dr. Will Frank, Bonn.

[Figure]

The author of this note helped to prepare the legal arguments for Luciano Lliuya's lawsuit against RWE because of the endangerment of the plaintiff's home by a possible glacial outburst flood risk (GLOF) (see Frank, The Huaraz Case, Climate Law Blog, Sabin Center, Columbia Law School, posted Dec. 17th, 2017; available at: http://blogs.law.columbia.edu/climatechange/2017/12/07/the-huaraz-case-lluiya-v-rwe-german-court-opens-recourse-to-climate-law-suit-against-big-co2-emitter/).

A) Summary of Facts

The authors of the study make the following statements, among others:

1) The water volume of Lake Palcacocha has increased significantly since the late 1970s (line 130). The water levels have developed as follows: Volume 1974: 3,690 x 10.000.000 m3; Volume 2009: 17,325 x 10.000.000 m3; Volume 2016: 17,403 x 10.000.000 m3. The water level was lowered by 3 m in 2018. 2) Palcacocha is one of the glacial lakes where glacial flooding can be triggered by rapid landslide processes (line 145). 3) Available studies agree that temperatures in the Peruvian Andes, including the Cordillera Blanca, have risen since the 1960s at a rate of about 0.2 to 0.3°C per decade and about 0.1°C per decade over the last 30 years (line 175). 4) The anthropogenic greenhouse effect has been identified as the most probable cause of warming over time (line178). 5) El Nino and la Nina and ENSO cannot explain the melting of glaciers over time (lines186-187). 6) According to various studies, climate change clearly plays a significant role in the melting of glaciers. This assessment is in line with the IPCC, according to which the melting of the glaciers in the Andes is very likely ("very high confidence") to be attributable to climate change (line 190). 7) According to a study from 2014, two thirds of the loss of mass of the glaciers in the Andes can be explained by global warming (line 192). 8) Almost 100% of the increase in the water volume of Lake Palcacocha is due to the melting of the glacier (line 199). 9) The growth of Lake Palcacocha has a clear anthropogenic component ("signal") (line 204).

B) Remarks

The authors come on the basis of the aforementioned facts to the conclusion:

"It can be said with certainty that without anthropogenic climate change the danger (of a GLOF) would be much lower, primarily because the size of the lake would then be substantially smaller and a long, flat glacier tongue would significantly weaken the energy of a possible rock or glacier retreat" (lines 225 - 228).

Despite the strong evidence for the causal link between the $CO_2$ emissions of RWE and the endangerment of the plaintiff's property the authors however find that

"at the current state of science an assessment of GLOF hazard attribution to anthropogenic climate change can only be qualitative." (lines 224f) and

"...the company's (= RWE) contribution to Palcacocha GLOF risk is proportionally small and hardly quantifiable." (lines 475 f).

The study by Huggel (et al.) thus doubts the quantifiability of the causal contribution of $CO_2$-emissions from RWE to the endangerment of the plaintiff's property. This assessment has possibly two reasons:

The authors of the study also include socio-economic factors as risk factors in their analysis, which in fact can hardly be quantified. These socio-economic aspects are however legally irrelevant to the question of whether a scientifically verifiable causal relationship exists between the emission of $CO_2$ by certain emitters and the risk of a GLOF. Possible failure of local government agencies to prevent damage does not rule out the primary responsibility of those causing the risk. Similarly, a deliberate acceptance of the risk of a GLOF by parts of the population would not preclude such liability of the causers of the risk, but would at most be of relevance in reducing their liability under the aspect of contributory negligence.

The second reason for the cautious conclusion of the authors of the study with respect to the quantifiability of the causal contribution of $CO_2$-emissions by big emitters to the GLOF-risk endangering Huaraz is probably to be seen in the fact that there are not yet

available detailed studies about the regional impacts of climate change in the Andes.

The problem regarding the possible regionalization of the impacts of climate change in the Cordillera Blanca does not, however, exclude the possibility that, even on the basis of current knowledge, a certain range of the size of the causal contribution of climate change to glacial melting in the area of Huaraz can be established.

If it is correct that

- RWE's CO2 emissions contribute in the magnitude of X percent to global warming, - according to the IPCC, the melting of the glaciers in the Andes is very likely to be due to climate change or, according to another study, at least two thirds of the loss of mass of the glacier in the Andes can be explained by global warming, - the rise in temperature in the region where the Palcacocha glacier is located is due to global warming as observed in the recent decades, - the increase in the water volume of the lake is almost 100% due to the melting of the Palcacocha glacier due to global warming, - the GLOF risk for Huaraz is due to the increased water volume of Lake Palcacocha,

then the causal contribution of RWE emissions to the GLOF risk lies between (at most) X percent (RWE's share in the increased pollution of the atmosphere with greenhouse gases) and a percentage of (at least) Y percent taking the remaining uncertainties with respect to the regionalization of global warming to the area of the Cordillera Blanca into account. Example: Assuming that possibly only two thirds of the mass loss of the Palcacocha Glacier can be traced with certainty to global warming because of still existing uncertainties with respect to regional effects of climate warming in the Andes Y (the causal contribution of CO2 emissions by RWE) would be 2/3 X.

This means: Even if it is not yet possible, according to the current state of knowledge, to precisely quantify the causal contribution of individual major emitters of CO2 to certain consequences of climate change, it does seem possible to determine a range within which this causal contribution must lie and thereby determine the minimum of the respective causal contribution to a certain impact of climate change.

C) Legal aspects

Even a "proportionally small" contribution to the GLOF-risk in question is not legally insignificant in view of the magnitude of the damage threatened by the GLOF.

The lawsuit filed against RWE is based on German law. As in other legal systems, the principle applies that the plaintiff must prove the facts on which his claim is based.

However, this principle does not apply unconditionally. In the case of damage being caused by multiple parties - as in the case of climate change being caused by a large number of actors - the court can estimate the causal contribution of individual parties.

Thus, even if the causal contribution of $CO_2$ emissions from RWE power plants cannot (yet) be exactly quantified in view of factors that may play a role in the causal chain it can – as the study of Huggel (et al.) confirms – be quantified to a degree that allows a reasonable estimation on which a judicial verdict on the responsibility of individual big $CO_2$ emitters can be based.

---

## Referee Comment (RC2) · Marcus Nüsser (Referee) · 22 Apr 2020

Marcus Nüsser marcus.nuesser@uni-heidelberg.de

Review

[Figure]

Based on the prominent case study of Lake Palcacocha in the Peruvian Andes the paper deals with the complexity of "socio-economical, institutional and cultural processes" which become drivers of risk exposure and vulnerability and ultimately shape glacier lake outburst flood risk for the urban agglomeration of Huaraz. Questions of associated responsibilities, causality, and justice in the context of the adverse effects of climate change are also raised. This integration of a "normative responsibility framework" is rather unusual and innovative in scientific studies on GLOFs. It extends the research perspective and integrates dimensions of political ecology and glaciology.

The paper is very well structured and the line of argument is convincing throughout the text. The introduction presents a clear statement of the problem with relevant references. When the authors mention the "impacts of glacier changes on natural and human systems" (l 32) they might also refer to papers on socio-hydrological interactions in other mountain regions in order to strengthen the global perspective. There are a number of contributions from various parts of the Himalaya. Such references would also be useful when the authors refer to adaptation strategies to cope with cryosphere changes (l 40-44). Socio-hydrological case studies dealing with meltwater dependent irrigated agriculture may be useful in this context (e.g. from Ladakh). The authors present their general understanding of this glacier riskscape "as a function of physical hazard, human exposure, and vulnerability of people and assets" (l 60), taking into consideration the IPCC-based framework and classical risk concepts developed by Blaikie, Oliver-Smith, and Wisner. The case of lake Palcacocha is particularly interesting as it is not only a representative case study for cryosphere risks (in the sense of the cryoscape) but it also presents a legal case where different actors are involved, in this case, a local Peruvian citizen and a German energy producer.

The case study of Lake Palcacocha is presented in a detailed and historically informed way. The physical and socio-economic drivers of risk are also presented in a convincing way. Section 4 provides important information on the socio-economic drivers of risk, especially the importance of recent urbanization processes, demographic growth,

and the role of different actor groups (Quechua-speaking farmers, mine workers, ruling Spanish-speaking classes) together with some remarks on settlement history. It is important to focus on class division, social marginalization and other economic factors to understand decision-making in urban planning and practices of building in hazard-prone regions. The authors might also add some examples from other mountain regions in the Global South to have a wider and more global context (people opt for habitat locations in hazard-prone flood plains because of economic gains). The last few lines of this section (l 299-300) are quite general (global histories of colonial power, neoliberalism, resource extraction). These aspects might be contextualized with social processes in the context of GLOF disasters. Section 5 provides detailed information on weak institutional structures and problems in regional governance over the past decades. The set of culturally embedded explanations of the 1941 GLOF and the important aspect of local concepts of place attachment are contextualized as other dimensions of explanation. The authors conclude that risk and associated loss and damage "is a multi-faceted construction and the question of causality can often not be fully solved, at least not in a quantitative way" (l 485-486). The paper uses a local case study to provide a multi-dimensional analysis with very relevant implications for international climate policy. This is innovative and deserves publication. The figures are relevant and illustrative.

Some few typos need to be corrected, e.g. l 104 two punctuations l 115 Hans Kinzl l 136 delete one bracket

---

## Short Comment (SC2) · 4 May 2020

Contribution to the Peer Review of "Anthropogenic climate change and glacier lake outburst flood risk: local and global drivers and responsibilities for the case of Lake Palcacocha, Peru" by Christian Huggel et al.

Rupert Stuart-Smith, Friederike Otto & Sihan Li, Environmental Change Institute, University of Oxford

Gerard Roe, Department of Earth and Space Sciences, University of Washington

[Figure]

Summary

The submitted manuscript aims to evaluate the physical and socioeconomic drivers of glacial lake outburst flood (GLOF) risk from Lake Palcacocha, a moraine-dammed proglacial lake in the Peruvian Andes. Based on a review of climate science and glaciology literature, the paper finds that 'the growth of lake Palcacocha has a clear anthropogenic signal' and that this has significantly increased the GLOF hazard. The paper also considers socio-economic, institutional, governance-related, cultural and emotional drivers of GLOF risk, and the resultant implications for greenhouse gas emitters' responsibility for compensatory payments in ongoing legal proceedings.

Our comments relate to the accuracy of (1) the authors' assessment of the attribution of GLOF risk to anthropogenic greenhouse gas emissions, and (2) the framework developed for understanding the drivers of GLOF risk in a legal context.

1. Accuracy of the GLOF hazard attribution

The authors note that (lines 191-203):

'The only available quantitative glacier attribution study that also includes the tropical Andes concludes that globally more than two thirds of the 1991-2010 glacier mass loss is due to anthropogenic forcing, and for tropical regions finds that an anthropogenic signal in observed glacier mass loss of recent decades is detectable with high confidence (Marzeion et al., 2014). The anthropogenic signal is much stronger for the past 2-3 decades as compared to earlier time periods. . . . Lake growth was highest in the 1990s and 2000s (Fig. 3), coinciding with the period where glacier shrinkage can regionally be attributed to anthropogenic emissions with high confidence. We therefore conclude that growth of lake Palcacocha has a clear anthropogenic signal.'

We are concerned that this statement underestimates the role of human influence on the GLOF hazard from Lake Palcacocha. Regional warming of 0.75-1.5 °C has been attributed to anthropogenic influence (Allen et al., 2018, see fig. 1.3) and mountain

glacier lengths act as lowpass filters of the climate they experience (Oerlemans, 2000; Roe, Baker and Herla, 2017) with multidecadal response times (Jóhannesson, Raymond and Waddington, 1989). This suggests that the expansion of Lake Palcacocha (and the resultant increase in GLOF hazard) is likely to have been attributable to human influence considerably prior to the period over which the authors indicate the presence of an anthropogenic signal.

Further to this, we are concerned the submitted manuscript misinterprets the findings of Marzeion et al., (2014). Taken at face value, the Marzeion et al. calculations would indicate that for low-latitude glaciers (shown in their supplementary material) anthropogenic influence on mass balance emerges only in the last three decades. However, (a) there would be a time lag of at least a decade between the emergence of human influence on mass balance and human influence on glacier lengths (Jóhannesson, Raymond and Waddington, 1989), (b) the glacier's length response to changes in mass balance is strongly influenced by the glacier topography and is therefore specific to the landscape on which the glacier lies, and (c) the results of Marzeion et al. (2014), if taken at face value, would indicate that mass balance was strongly negative throughout the 20th Century. Therefore, if they were an accurate representation of Palcaraju glacier's mass balance, significant glacial retreat would be expected throughout the historical period, rather than only emerging in recent decades as has been observed (as described by the Huggel et al. manuscript). Consequently, the concurrence between the period of time for which mass loss is most attributable to anthropogenic greenhouse gas emissions (according to Marzeion et al., 2014) and the period of most rapid expansion of Lake Palcacocha does not appear to be a solid foundation on which to make a statement of the role of human influence on the observed retreat of this glacier.

With respect to the effect of Lake Palcacocha's expansion on the GLOF hazard, the authors explain that (lines 210-213):

"Some of the factors [influencing GLOF magnitude and probability] (such as lake formation) are closely related to climate change while others can be associated to geologic

or geotechnical conditions (e.g. dam stability), or are explicitly influenced by human intervention aiming at reducing the risk of GLOFs (e.g. lake freeboard determined by the height of the constructed drainage canal)."

Anthropogenically-driven glacial retreat has directly resulted in the formation of a large proglacial lake and is the primary and necessary cause of GLOF risk. Non-climatic factors mediate the impact of climate change on the GLOF hazard, but in our view the authors should state explicitly that these factors are less important causes of the present GLOF hazard. The need to implement adaptive measures to reduce GLOF risk is therefore the result of the expansion of Lake Palcacocha and the other factors identified by the authors are largely incidental and would have been inconsequential but for the impact of climate change on the glacier's retreat.

2. Accuracy of the framework for understanding risk, and its relevance for ongoing legal proceedings

The paper gives comparable weight to the physical and 'interacting socio-economic, institutional and cultural processes' which contribute to flood risk. The existence of these social influences on flood risk is not challenged here but this framing obscures the fact that present GLOF risk is a direct result of the anthropogenically-driven retreat of the glacier.

Huggel et al present 'risk (and associated loss and damage) [as] a multi-faceted construction' and argue that 'the question of causality can often not be fully solved, at least not in a quantitative way' and 'in contrast [to the physical GLOF risk causal chain] the current conditions of exposure and vulnerability of people and values in Huaraz to GLOF hazard can only be understood with a historical perspective of social, economic, political and cultural dynamics'. It is undoubtedly true that exposure and vulnerability play a crucial role in determining the ultimate risk, however, non-climatic factors, such as the decision of Spanish colonialists to select this location for the city of Huaraz in the 16th Century (etc) would not have mattered if not for climate change and thus are,

in our view, legally irrelevant as far as the question of causality is concerned. Similarly, we believe that the possible failure of local government agencies to develop successful risk-reduction mechanisms is not relevant for understanding the primary cause of the present GLOF hazard and therefore need to implement costly adaptation measures. Climate change is a necessary cause of flood risk in this setting.

References

Allen, M. R. et al. (2018) 'Framing and Context', in Masson-Delmotte, V. et al. (eds) Global Warming of 1.5°C. An IPCC Special Report on the impacts of global warming of 1.5°C above pre-industrial levels and related global greenhouse gas emission pathways, in the context of strengthening the global response to the threat of climate change, pp. 49–91.

Jóhannesson, T., Raymond, C. and Waddington, E. (1989) 'Time–Scale for Adjustment of Glaciers to Changes in Mass Balance', Journal of Glaciology, 35(121), pp. 355–369. doi: 10.3189/S002214300000928X.

Marzeion, B. et al. (2014) 'Attribution of global glacier mass loss to anthropogenic and natural causes', Science. American Association for the Advancement of Science (AAAS), 345(6199), pp. 919–921. doi: 10.1126/science.1254702.

Oerlemans, J. (2000) 'Holocene glacier fluctuations: is the current rate of retreat exceptional?', Annals of Glaciology, 31, pp. 39–44. doi: 10.3189/172756400781820246.

Roe, G. H., Baker, M. B. and Herla, F. (2017) 'Centennial glacier retreat as categorical evidence of regional climate change', Nature Geoscience. Nature Publishing Group, 10(2), pp. 95–99. doi: 10.1038/ngeo2863.

---

## Author Comment (AC1) · 5 Jun 2020

We generally would like to express to acknowledge the review comments and suggestions made by the two reviewers and the additional comments contributed by colleagues. We appreciate the time our colleagues have taken to read and analyze the paper, especially during the current difficult times. We found all comments helpful and suggestions constructive and are very positive that we can appropriately address all points.

[Figure]

Response to review comments by Dmitry Petrakov

General comments: The Lake Palcacocha was a source of the most destructive GLOF in human history which happened in 1941. Some mitigation measures to prevent lake outburst have been done. Growing volume of the lake due to glacier retreat is accompanying by growing population of Huaraz city located at the flow path. Thus any attempts to assess risk of GLOF from the Lake Palcacocha are highly relevant both at local and global scale. Authors used the Lake Palcacocha as a representative case for other glacier lakes and related risks around the world. The paper provides high-quality case study with significant conclusions, both locally and globally. It also provides nice synthesis of natural and social sciences which is important for comprehensive risk assessments. Novelty of results is quite clear. Structure of the paper is reasonable, Introduction is well-written, motivation and goals of the paper are fully clear. Authors provide brief but comprehensive description of the Lake Palcacocha evolution and engineering solutions to prevent dam failure. Currently lake growth potential is limited by topographic constraints, but outburst probability is high. Authors analyzed anthropogenic contribution to glacier retreat in Cordillera Blanca based on literature review and concluded that growth of lake Palcacocha has a clear anthropogenic signal. GLOF hazard in the absence of anthropogenic climate change the flood hazard would be much lower due to change of lake volume and increasing impact energy of ice/rock avalanches. Socio-economic drivers of risk are determined and analyzed basing on literature review and survey conducted in 2017. What is important, local residents showed little concern for the risk of flooding. The institutional instability generated only short-term, unsustainable measures to protect downstream populations. Authors note that combined effects of institutional and governance-related risk drivers have contributed to the increase of risk and provide interesting insight on cultural and emotional component of risk. Basing on analysis of risk drivers authors provide implications for responsibility and justice. The manuscript is well written and free of technical errors, well structured, appropriate in length. All figures and the table are high-quality. Supplementary material is very useful for understanding of hazard assessment technology. The conclusions

are clear and precise. The results obtained in this study are highly relevant to assess risk of future GLOFs not just in Huaraz but elsewhere. I definitely support publication of the manuscript.

Author response: Thanks for this analysis of the study and appreciation of our work.

Specific comments: Authors noted that previous studies estimated about 40,000 people living in the inundation zone with a potential death toll of close to 20,000 (Somos-Valenzuela, 2014). Anywhere there is no assessment of current situation despite Fig.7 demonstrates significant growth of population in Huaraz. How many people live in zones with different hazard level (Fig.5) now? Is population density within hazardous area uniform or not? An addition of a figure where population density will be provided solely or overlaid to hazard zonation might be interesting for readers and useful for local communities and decision makers. Being fully agree with author's concept I will be happy to see what components of risk are major and what components are really minor. Furthermore, some recommendation how to minimize GLOF risk in Huaraz basing on risk driver analysis will increase practical and intellectual merit of this really great paper.

Author response: Thanks for pointing out these aspects. We were investigating digital data on population distribution in Huaraz, and thanks to our colleague Marcelo Somos-Valenzuela were able to get access to pertinent GIS based data which allowed us to perform additional analysis to address Dmitry Petrakov's comments and suggestions. This analysis which is supported by a new figure (see below) indicates that high population density intersects with the high hazard zone. We carefully evaluated how we could best graphically represent the different information layers in one figure such that the reader could quickly infer hazard zones vs population density. Our graphic analysis eventually concluded that it is not sensible to bring all information layers together in one single figure and we therefore decide on two panels which will be an extension of the current Figure 5 (see below). However, in terms of risks we also add here a word of caution: population data only refers to residential population, but the GLOF risk

threatens parts of the city center, including markets, touristic attractions, bus stations, etc., which can have much higher concentration of people, depending on daytime. And eventually, the death toll is very difficult to estimate because death depends on multiple factors such as detailed structural building data, physical conditions of people depending on additional factors such as age, health conditions, daytime (see above) etc. We don't think we're in a position to perform such as estimate and therefore mention the estimate given by Somos-Valenzuela (2014).

We include results of all this additional analysis in section 3 of the paper (physical drivers of risk).

Figure 1: New figure showing the population density in Huaraz, to go in tandem with the original Figure 5.

Figure 2: For reference, Figure 5 from the original manuscript.

The point about an evaluation of major and minor components of risk is a very interesting one but at the same time also very challenging to perform in the absence of a fully quantitative risk analysis across all different risk drivers. It is important to consider that risk is a function of hazard, exposure and vulnerability, and hence, if there is no hazard or no exposure no risk exists. So one could argue that without the existence of Lake Palcacocha (or other glacier lakes above Huaraz including Shallap, Cuchillacocha, Tulparaju, and Llaca) no GLOF risk in Huaraz would exist. Likewise if there were no exposed assets such as homes, infrastructure, or agriculture, then no (or minimal) GLOF risk would exist. We think that the question of major vs minor risk components can only be appropriately addressed if different risk components are weighted, which, eventually is a societal or political question. For instance, is a building or a human life weighted higher? We would like to abstain from weighting different risk components, and hence we would not be in a position to make a statement about major/minor risk components. However, we think it is an important discussion and we therefore include this issue in section 7.

Concerning recommendations for risk reduction measures we now include an additional paragraph in section 7 which addresses this point in a comprehensive way and with reference to the different responsibilities.
* * *
[Figure]

**Fig. 1.**

**Hazard level**
residual
low
medium
high

Main square

Nueva Florida

N

| 0 | 0.5 | 1 | 1.5 | 2 km |

**Fig. 2.**

---

## Author Comment (AC4) · 5 Jun 2020

Author response: We would like to thank the author team of this comment for their engagement with this study and the very valuable comments they provide. To some extent similar as in the case of the comment by Will Frank we think that we're departing from somewhat different perspectives. In our paper we are primarily interested in how risk evolves over time and what the driving forces are from local to global, and what this means in terms of responsibilities. In this comment there is a stronger push towards the causality issue, which is also important, and we do touch upon it to some extent

as well in our paper, but not as a central aspect. Having said this, we're nevertheless glad that you bring up the issue of glacier (and glacier lake) attribution to anthropogenic climate change. Unfortunately, we're confronted with a scarcity of research on climate-glacier attribution in general, and in particular for the tropical Andes region, by far the largest glacierized region in the tropics. We built our attribution analysis (section 3) on existing literature from climatic change in the region to climate-glacier studies. We also agree that we should be more careful in referring to the Marzeion et al. (2014) paper and we clearly acknowledge the problems as you outline. We may add to this that one of the underlying problems of this study (and others as well) is the inadequate climate data that is used, with insufficient quality, or temporal and spatial coverage/resolution, a limitation that many climate datasets for this region share due to scarce climatological on the ground measurements and complex topography and climatology in the high Andes. While we don't see the space and scope in our paper to go into substantial detail about climate-glacier change attribution, we revised section 3 to accommodate the concerns that are pointed out here. Specifically, we are happy to take on board the new evidence available from Stuart-Smith et al. (in review). Based on this comment and the previous one by Will Frank, we revised the following text sections: lines 191-194 (putting the Marzeion et al. 2014 study in perspective, with more evidence by Stuart-Smith et al. in review); lines 224-228: we revised the statement that GLOF hazard attribution to anthropogenic climate change can only be qualitative and refere here to the Stuart-Smith et al study.

Reference: Stuart-Smith, R.F., Roe, G.H., Li, S., Allen, M.R., 2020. Anthropogenic contribution to the retreat of Palcaraju glacier (Cordillera Blanca, Peru) and glacial lake outburst flood risk. Nat. Geosci. in review

Furthermore, based on the comments by Will Frank and Stuart-Smith et al. we also recognize the need for further clarification of the type of responsibilities we're addressing in our study. We do not focus on any legal responsibility. We added a clarification at the beginning of section 7 (Implications for responsibility and justice) where we first

outline the four aspects when assigning responsibilities. We specifically add that this conceptualization of responsibility encompasses aspects of legal liability, explaining the link between the subject of responsibility and the object of responsibility, but clarify that our understanding of responsibility goes beyond the legal and liability framing. This also addresses the concerns of the two comments that the social, economic and institutional drivers of risk are irrelevant for the court case or decision. We basically agree on this (but the court decision will give us more precision in this respect) but for our concept of responsibility these risk drivers are clearly relevant.

---

## Author Response (AR1)

**Response to review comments on Huggel et al: Anthropogenic climate change and glacier lake outburst flood risk: local and global drivers and responsibilities for the case of Lake Palcacocha, Peru.**

*We generally would like to express to acknowledge the review comments and suggestions made by the two reviewers and the additional comments contributed by colleagues. We appreciate the time our colleagues have taken to read and analyze the paper, especially during the current difficult times. We found all comments helpful and suggestions constructive and are very positive that we can appropriately address all points.*
*In the following, review comments are in normal font and our response is in italic.*

**Response to review comments by Dmitry Petrakov**

General comments: The Lake Palcacocha was a source of the most destructive GLOF in human history which happened in 1941. Some mitigation measures to prevent lake outburst have been done. Growing volume of the lake due to glacier retreat is accompanying by growing population of Huaraz city located at the flow path. Thus any attempts to assess risk of GLOF from the Lake Palcacocha are highly relevant both at local and global scale. Authors used the Lake Palcacocha as a representative case for other glacier lakes and related risks around the world. The paper provides high-quality case study with significant conclusions, both locally and globally. It also provides nice synthesis of natural and social sciences which is important for comprehensive risk assessments.
Novelty of results is quite clear. Structure of the paper is reasonable, Introduction is well-written, motivation and goals of the paper are fully clear. Authors provide brief but comprehensive description of the Lake Palcacocha evolution and engineering solutions to prevent dam failure. Currently lake growth potential is limited by topographic constraints, but outburst probability is high. Authors analyzed anthropogenic contribution to glacier retreat in Cordillera Blanca based on literature review and concluded that growth of lake Palcacocha has a clear anthropogenic signal. GLOF hazard in the absence of anthropogenic climate change the flood hazard would be much lower due to change of lake volume and increasing impact energy of ice/rock avalanches. Socio-economic drivers of risk are determined and analyzed basing on literature review and survey conducted in 2017. What is important, local residents showed little concern for the risk of flooding. The institutional instability generated only short-term, unsustainable measures to protect downstream populations. Authors note that combined effects of institutional and governance-related risk drivers have contributed to the increase of risk and provide interesting insight on cultural and emotional component of risk. Basing on analysis of risk drivers authors provide implications for responsibility and justice.
The manuscript is well written and free of technical errors, well structured, appropriate in length. All figures and the table are high-quality. Supplementary material is very useful for understanding of hazard assessment technology. The conclusions are clear and precise. The results obtained in this study are highly relevant to assess risk of future GLOFs not just in Huaraz but elsewhere. I definitely support publication of the manuscript.

*Author response:*
*Thanks for this analysis of the study and appreciation of our work.*

Specific comments:
Authors noted that previous studies estimated about 40,000 people living in the inundation zone with a potential death toll of close to 20,000 (Somos-Valenzuela, 2014).
Anywhere there is no assessment of current situation despite Fig.7 demonstrates significant growth of population in Huaraz. How many people live in zones with different hazard level (Fig.5) now? Is population density within hazardous area uniform or not? An addition of a figure where population density will be provided solely or overlaid to hazard zonation might be interesting for readers and useful for local communities and decision makers. Being fully agree with author's concept I will be happy to see what components of risk are major and what components are really minor. Furthermore, some recommendation how to minimize GLOF risk in Huaraz basing on risk driver analysis will increase practical and intellectual merit of this really great paper.

*Thanks for pointing out these aspects.*

*We were investigating digital data on population distribution in Huaraz, and thanks to our colleague Marcelo Somos-Valenzuela were able to get access to pertinent GIS based data from the National Statistical Institute INEI which allowed us to perform additional analysis to address Dmitry Petrakov's comments and suggestions. This analysis which is supported by a new figure indicates that high population density intersects with the high hazard zone. We carefully evaluated how we could best graphically represent the different information layers in one figure such that the reader could quickly infer hazard zones vs population density. A full overlay of population distribution data, with hazard zones and underlying satellite data is not feasible, therefore, for reasons of visual clarity we include now two panels in Figure 5. Nevertheless, we indicate the extent of the high hazard zone in the population distribution map such that the reader can quickly grasp the relevant information.*

*However, in terms of risks we also add here a word of caution: population data only refers to residential population, but the GLOF risk threatens parts of the city center, including markets, business centers,, touristic attractions, bus stations, etc., which can have much higher concentration of people, depending on daytime. And eventually, the death toll is very difficult to estimate because death depends on multiple factors such as detailed structural building data, physical conditions of people depending on additional factors such as age, health conditions, daytime (see above) etc. We don't think we're in a position to perform such as estimate of death toll and therefore mention the estimate given by Somos-Valenzuela (2014).*

*We include results of all this additional analysis in section 3 of the paper (physical drivers of risk), and the revised text section is now as follows:*

*"Previous studies estimated about 40,000 people living in the inundation zone with a potential death toll of close to 20,000 (Somos-Valenzuela, 2014). Based on spatial census data from the National Statistical Institute of Peru (INEI), here we found that about 22,500 inhabitants living in the high hazard zone are highly exposed to GLOF (Fig. 5). However, because the high hazard zone intersects with the central business and market places of Huaraz the number of people present during the day times is much higher, possibly up to 50,000."*

*Additionally, in the Supplementary Material, we slightly revised the text explaining in further detail the method to generate the hazard zones in Huaraz. We specify that we use a threshold of 1 m inundation height to distinguish between medium and high intensity zones which are then converted into hazard zones in line with international standards.*

*The point about an evaluation of major and minor components of risk is a very interesting one but at the same time also very challenging to perform in the absence of a fully quantitative risk analysis across all different risk drivers. It is important to consider that risk is a function of hazard, exposure and vulnerability, and hence, if there is no hazard or no exposure no risk exists. So one could argue that without the existence of Lake Palcacocha (or other glacier lakes above Huaraz including Shallap, Cuchillacocha, Tulparaju, and Llaca) no GLOF risk in Huaraz would exist. Likewise if there were no exposed assets such as people, homes, infrastructure, or agriculture, then no (or minimal) GLOF risk would exist. We think that the question of major vs minor risk components can only be appropriately*

*addressed if different risk components are weighted, which, eventually is a societal or political question. For instance, is a building or a human life weighted higher? We would like to abstain from weighting different risk components, and hence we would not be in a position to make a statement about major/minor risk components. However, we think it is an important discussion and we therefore include this issue in section 7.*

*Concerning recommendations for risk reduction measures we mention quite many different types of measures in section 7 when referring to different responsibilities but we now include also a short additional text section with related recommendations in section 8.*

**Response to review comments by Marcus Nüsser**

Reviewer comment:
Based on the prominent case study of Lake Palcacocha in the Peruvian Andes the paper deals with the complexity of "socio-economical, institutional and cultural processes" which become drivers of risk exposure and vulnerability and ultimately shape glacier lake outburst flood risk for the urban agglomeration of Huaraz. Questions of associated responsibilities, causality, and justice in the context of the adverse effects of climate change are also raised. This integration of a "normative responsibility framework" is rather unusual and innovative in scientific studies on GLOFs. It extends the research perspective and integrates dimensions of political ecology and glaciology. The paper is very well structured and the line of argument is convincing throughout the text. The introduction presents a clear statement of the problem with relevant references.

*Author response:*
*Thanks for this analysis.*

Reviewer comments:
When the authors mention the "impacts of glacier changes on natural and human systems" (l 32) they might also refer to papers on socio-hydrological interactions in other mountain regions in order to strengthen the global perspective. There are a number of contributions from various parts of the Himalaya. Such references would also be useful when the authors refer to adaptation strategies to cope with cryosphere changes (l 40-44). Socio-hydrological case studies dealing with meltwater dependent irrigated agriculture may be useful in this context (e.g. from Ladakh).

*Author response:*
*Yes, good point, we now include additional reference to studies of socio-hydrological interactions in the Introduction section of the paper, specifically we include several references with additional indications on international (incl Himalayas and Ladakh studies) studies on socio-hydrologic and socio-cryospheric research and emphasize the important progress that has been made in this field in recent years. We include the following additional references (included in the paragraph of lines 38-48).*

*Allison, E. A.: The spiritual significance of glaciers in an age of climate change, WIREs Climate Change, 6, 493-508, 2015.*

*Carey, M., McDowell, G., Huggel, C., Jackson, J., Portocarrero, C., Reynolds, J. M., and Vicuña, L.: Integrated Approaches to Adaptation and Disaster Risk Reduction in Dynamic Socio-cryospheric Systems, in: Snow and Ice-Related Hazards, Risks, and Disasters, edited by: Haeberli, W., and Whiteman, C., Elsevier, Amsterdam, 219-261, 2014.*

Drew, G.: A Retreating Goddess? Conflicting Perceptions of Ecological Change near the Gangotri-Gaumukh Glacier, Journal for the Study of Religion, Nature and Culture, 6, 344-362, 2012.

Gagné, K.: Caring for Glaciers: Land, Animals, and Humanity in the Himalayas, University of Washington Press, Seattle, 2019.

Gagné, K., Rasmussen, M. B., and Orlove, B.: Glaciers and Society: Attributions, Perceptions, and Valuations, WIREs Climate Change, 5, 793-808, 2014.

Mukherji, A., Sinisalo, A., Nüsser, M., Garrard, R., and Eriksson, M.: Contributions of the cryosphere to mountain communities in the Hindu Kush Himalaya: a review, Regional Environmental Change, 19, 1311-1326, 2019.

Nüsser, M., and Baghel, R.: The Emergence of the Cryoscape: Contested Narratives of Himalayan Glacier Dynamics and Climate Change, in: Environmental and Climate Change in South and Southeast Asia, edited by: Schuler, B., Koninklijke Brill, Leiden, 138-156, 2014.

Nüsser, M., and Baghel, R.: Local Knowledge and Global Concerns: Artificial Glaciers as a Focus of Environmental Knowledge and Development Interventions, in: Ethnic and Cultural Dimensions of Knowledge. Knowledge and Space, edited by: Neusburger, P., Freytag, T., and Suarsana, L., Springer, Switzerland, 191-209, 2016.

Nüsser, M., Dame, J., Kraus, B., Baghel, R., and Schmidt, S.: Socio-hydrology of "artificial glaciers" in Ladakh, India: assessing adaptive strategies in a changing cryosphere, Regional Environmental Change, 19, 1327-1337, 2019.

Orlove, B., Milch, K., Zaval, L., Ungemach, C., Brugger, J., Dunbar, K., and Jurt, C.: Framing climate change in frontline communities: anthropological insights on how mountain dwellers in the USA, Peru, and Italy adapt to glacier retreat, Regional Environmental Change, 19, 1295-1309, 2019.

Sherry, J., Curtis, A., Mendham, E., and Toman, E.: Cultural landscapes at risk: Exploring the meaning of place in a sacred valley of Nepal, Global Environmental Change, 52, 190-200, 2018.

Sörlin, S.: Cryo-History: Narratives of Ice and the Emerging Arctic Humanities, in: The New Arctic, edited by: Evengård, B., Larsen, J. N., and Paasche, Ø., Springer, New York, 327-339, 2015.

Williams, C., and Golovnev, I.: Pamiri Women and the Melting Glaciers of Tajikistan, in: A Political Ecology of Women, Water and Global Environmental Change, edited by: Buechler, S., and Hanson, A.-M. S., Routledge, New York, chapter 11, 2015.

Reviewer comment:
The authors present their general understanding of this glacier riskscape "as a function of physical hazard, human exposure, and vulnerability of people and assets" (l 60), taking into consideration the IPCC-based framework and classical risk concepts developed by Blaikie, Oliver-Smith, and Wisner. The case of lake Palcacocha is particularly interesting as it is not only a representative case study for cryosphere risks (in the sense of the cryoscape) but it also presents a legal case where different actors are involved, in this case, a local Peruvian citizen and a German energy producer.
The case study of Lake Palcacocha is presented in a detailed and historically informed way. The physical and socio-economic drivers of risk are also presented in a convincing way. Section 4 provides important information on the socio-economic drivers of risk, especially the importance of recent urbanization processes, demographic growth, and the role of different actor groups (Quechua-speaking farmers, mine workers, ruling Spanish-speaking classes) together with some remarks on settlement history. It is important to focus on class division, social marginalization and other economic factors to understand decision-making in urban planning and practices of building in hazard prone regions. The authors might also add some examples from other mountain regions in the Global South to have a wider and more global context (people opt for habitat locations in hazard-prone flood plains because of economic gains).

*Author response:*
*Agreed, we now put the Palcacocha case in the context of other, potentially similar cases in developing countries, specifically, we make reference to cases, including in the Himalayas but also in other parts of the Andes, North America and Europe. We add a paragraph addressing this comment in section 6, on the cultural components of risk.*

*"These trends in the Cordillera Blanca also exist internationally, and people knowingly inhabit areas exposed to GLOFs in other glacier-fed watersheds. In some cases, they are "forced" into these areas due to cheaper land in the floodplain or nearby job and livelihood opportunities (Carey et al., 2014; Orlove et al., 2019). In other cases, they select GLOF-prone sites to live due to historical and cultural connections to those flood-prone places (Sherry et al., 2018), or they utilize other cultural or spiritual techniques to manage glacier-related risks (Allison, 2015; Gagné, 2019), or they possess different local knowledge about risk that sometimes differs from scientific or institutional assessments of GLOF risks (Drew, 2012; Williams and Golovnev, 2015). Furthermore, in India for example, there are also documented recent major GLOF disasters due to exposure and high vulnerability of a large number of people due to religious and tourism related reasons (Allen et al., 2016)."*

Reviewer comment:
The last few lines of this section (l 299-300) are quite general (global histories of colonial power, neoliberalism, resource extraction). These aspects might be contextualized with social processes in the context of GLOF disasters.

*Author response:*
*These two lines are intentionally quite general because these lines are at the end of the respective section 4 and they should summarize the points made in the above text of this section. But we believe to see Marcus Nüsser's point and now modified the text at the end of this section to make more specific reference to the conditions in Huaraz.*

Reviewer comment:
Section 5 provides detailed information on weak institutional structures and problems in regional governance over the past decades. The set of culturally embedded explanations of the 1941 GLOF and the important aspect of local concepts of place attachment are contextualized as other dimensions of explanation. The authors conclude that risk and associated loss and damage "is a multi-faceted construction and the question of causality can often not be fully solved, at least not in a quantitative way" (l 485-486). The paper uses a local case study to provide a multi-dimensional analysis with very relevant implications for international climate policy. This is innovative and deserves publication. The figures are relevant and illustrative.

*Author response:*
*Comments and evaluation of our study appreciated.*

Reviewer comment:
Some few typos need to be corrected, e.g. l 104 two punctuations l 115 Hans Kinzl l delete one bracket.

*Author response:*
*Thanks, we corrected these typos.*

**Response to short comment by Will Frank**

The author of this note helped to prepare the legal arguments for Luciano Lliuya's lawsuit against RWE because of the endangerment of the plaintiff's home by a possible glacial outburst flood risk (GLOF) (see Frank, The Huaraz Case, Climate Law Blog, Sabin Center, Columbia Law School, posted Dec. 17th, 2017; available at: http://blogs.law.columbia.edu/climatechange/2017/12/07/the-huaraz-case-lluiya-v- rwe-german-court-opens-recourse-to-climate-law-suit-against-big-co2-emitter/).

A) Summary of Facts

The authors of the study make the following statements, among others:

1) The water volume of Lake Palcacocha has increased significantly since the late 1970s (line 130). The water levels have developed as follows: Volume 1974: 3,690 x 10.000.000 m3; Volume 2009: 17,325 x 10.000.000 m3; Volume 2016: 17,403 x 10.000.000 m3. The water level was lowered by 3 m in 2018. 2) Palcacocha is one of the glacial lakes where glacial flooding can be triggered by rapid landslide processes (line 145). 3) Available studies agree that temperatures in the Peruvian Andes, including the Cordillera Blanca, have risen since the 1960s at a rate of about 0.2 to 0.3$^\circ$C per decade and about 0.1$^\circ$C per decade over the last 30 years (line 175). 4) The anthropogenic greenhouse effect has been identified as the most probable cause of warming over time (line178). 5) El Nino and la Nina and ENSO cannot explain the melting of glaciers over time (lines186-187). 6) According to various studies, climate change clearly plays a significant role in the melting of glaciers. This assessment is in line with the IPCC, according to which the melting of the glaciers in the Andes is very likely ("very high confidence") to be attributable to climate change (line 190). 7) According to a study from 2014, two thirds of the loss of mass of the glaciers in the Andes can be explained by global warming (line 192). 8) Almost 100% of the increase in the water volume of Lake Palcacocha is due to the melting of the glacier (line 199). 9) The growth of Lake Palcacocha has a clear anthropogenic component ("signal") (line 204).

B) Remarks

The authors come on the basis of the aforementioned facts to the conclusion:

"It can be said with certainty that without anthropogenic climate change the danger (of a GLOF) would be much lower, primarily because the size of the lake would then be substantially smaller and a long, flat glacier tongue would significantly weaken the energy of a possible rock or glacier retreat" (lines 225 - 228).

Despite the strong evidence for the causal link between the CO2 emissions of RWE and the endangerment of the plaintiff[t]s property the authors however find that "at the current state of science an assessment of GLOF hazard attribution to anthropogenic climate change can only be qualitative." (lines 224f) and "...the company's (= RWE) contribution to Palcacocha GLOF risk is proportionally small and hardly quantifiable." (lines 475 f).

The study by Huggel (et al.) thus doubts the quantifiability of the causal contribution of CO2-emissions from RWE to the endangerment of the plaintiff[t]s property. This assessment has possibly two reasons:

The authors of the study also include socio-economic factors as risk factors in their analysis, which in fact can hardly be quantified. These socio-economic aspects are however legally irrelevant to the question of whether a scientifically verifiable causal relationship exists between the emission of CO2 by certain emitters and the risk of a GLOF. Possible failure of local government agencies to prevent damage does not rule out the primary responsibility of those causing the risk. Similarly, a deliberate acceptance of the risk of a GLOF by parts of the population would not preclude such liability of the causers of the risk, but would at most be of relevance in reducing their liability under the aspect of contributory negligence.

The second reason for the cautious conclusion of the authors of the study with respect to the quantifiability of the causal contribution of CO2-emissions by big emitters to the GLOF-risk endangering Huaraz is probably to be seen in the fact that there are not yet available detailed studies about the regional impacts of climate change in the Andes.

The problem regarding the possible regionalization of the impacts of climate change in the Cordillera Blanca does not, however, exclude the possibility that, even on the basis of current knowledge, a certain range of the size of the causal contribution of climate change to glacial melting in the area of Huaraz can be established.

If it is correct that RWE's CO2 emissions contribute in the magnitude of X percent to global warming, - according to the IPCC, the melting of the glaciers in the Andes is very likely to be due to climate change or, according to another study, at least two thirds of the loss of mass of the glacier in the Andes can be explained by global warming, - the rise in temperature in the region where the Palcacocha glacier is located is due to global warming as observed in the recent decades, - the increase in the water volume of the lake is almost 100% due to the melting of the Palcacocha glacier due to global warming,
- the GLOF risk for Huaraz is due to the increased water volume of Lake Palcacocha, then the causal contribution of RWE emissions to the GLOF risk lies between (at most) X percent (RWE$^t$s share in the increased pollution of the atmosphere with greenhouse gases) and a percentage of (at least) Y percent taking the remaining uncertainties with respect to the regionalization of global warming to the area of the Cordillera Blanca into account. Example: Assuming that possibly only two thirds of the mass loss of the Palcacocha Glacier can be traced with certainty to global warming because of still existing uncertainties with respect to regional effects of climate warming in the Andes Y (the causal contribution of CO2 emissions by RWE) would be 2/3 X.

This means: Even if it is not yet possible, according to the current state of knowledge, to precisely quantify the causal contribution of individual major emitters of CO2 to certain consequences of climate change, it does seem possible to determine a range within which this causal contribution must lie and thereby determine the minimum of the respective causal contribution to a certain impact of climate change.

C) Legal aspects

Even a "proportionally small" contribution to the GLOF-risk in question is not legally insignificant in view of the magnitude of the damage threatened by the GLOF.

The lawsuit filed against RWE is based on German law. As in other legal systems, the principle applies that the plaintiff must prove the facts on which his claim is based.

However, this principle does not apply unconditionally. In the case of damage being caused by multiple parties - as in the case of climate change being caused by a large number of actors - the court can estimate the causal contribution of individual parties.

Thus, even if the causal contribution of CO2 emissions from RWE power plants cannot (yet) be exactly quantified in view of factors that may play a role in the causal chain it can – as the study of Huggel (et al.) confirms – be quantified to a degree that allows a reasonable estimation on which a judicial verdict on the responsibility of individual big CO2 emitters can be based.

*Author response:*
*We much appreciate this careful analysis provided by Will Frank. We also appreciate the transparency he provides in being involved in the RWE vs Saul Lliuya case. Accordingly, his analysis departs from ours*

*and offers a different perspective. Our intention is not to analyze issues specifically pertinent to the ongoing legal case, and hence our objective is not to perform an analysis of causality. Rather, our focus is on a comprehensive approach to analyze how multiple drivers of risk, dynamically and over time combine to result in the risk people of Huaraz currently face vis-à-vis a GLOF from Palcacocha. We connect this analysis with the discussion of different associated responsibilities, but again, do not intend to go into any legal responsibility or liability issues. To make the purpose and context of our study clearer we add a specification in the Introduction section (lines 99-100).*

*We agree that a causal contribution indicating a range is feasible.*

*In our revision we furthermore address the text section in section 7, lines 585-589, about the contribution of RWE to GLOF risk at Palcacocha and Huaraz. While we think that the basic content of the statement holds valid we recognize that some precision in wording is needed. As addressed in response to the comment below we can distinguish between GLOF hazard and risk, and related attribution, and the ability to quantify contributions by GHG emitters. In the revised version we refer to the potential (and limitations) of quantifying the contribution of global GHG emitters, possibly also indicating a range of contribution. However, we continue to put this in a context of broader risk responsibilities.*

**Response to short comment by Rupert Stuart-Smith, Fredi Otto, Sihan Li, Gerard Roe**

Summary

The submitted manuscript aims to evaluate the physical and socioeconomic drivers of glacial lake outburst flood (GLOF) risk from Lake Palcacocha, a moraine-dammed proglacial lake in the Peruvian Andes. Based on a review of climate science and glaciology literature, the paper finds that 'the growth of lake Palcacocha has a clear anthropogenic signal' and that this has significantly increased the GLOF hazard. The paper also considers socio-economic, institutional, governance-related, cultural and emotional drivers of GLOF risk, and the resultant implications for greenhouse gas emitters' responsibility for compensatory payments in ongoing legal proceedings.

Our comments relate to the accuracy of (1) the authors' assessment of the attribution of GLOF risk to anthropogenic greenhouse gas emissions, and (2) the framework developed for understanding the drivers of GLOF risk in a legal context.

1. Accuracy of the GLOF hazard attribution

The authors note that (lines 191-203):

'The only available quantitative glacier attribution study that also includes the tropical Andes concludes that globally more than two thirds of the 1991-2010 glacier mass loss is due to anthropogenic forcing, and for tropical regions finds that an anthropogenic signal in observed glacier mass loss of recent decades is detectable with high confidence (Marzeion et al., 2014). The anthropogenic signal is much stronger for the past 2-3 decades as compared to earlier time periods. Lake growth was highest in the

1990s and 2000s (Fig. 3), coinciding with the period where glacier shrinkage can regionally be attributed to anthropogenic emissions with high confidence. We therefore conclude that growth of lake Palcacocha has a clear anthropogenic signal.'

We are concerned that this statement underestimates the role of human influence on the GLOF hazard from Lake Palcacocha. Regional warming of 0.75-1.5 $^\circ$C has been attributed to anthropogenic influence (Allen et al., 2018, see fig. 1.3) and mountain glacier lengths act as lowpass filters of the climate they experience (Oerlemans, 2000; Roe, Baker and Herla, 2017) with multidecadal response times (Jóhannesson, Raymond and Waddington, 1989). This suggests that the expansion of Lake Palcacocha (and the resultant increase in GLOF hazard) is likely to have been attributable to human influence considerably prior to the period over which the authors indicate the presence of an anthropogenic signal.

Further to this, we are concerned the submitted manuscript misinterprets the findings of Marzeion et al., (2014). Taken at face value, the Marzeion et al. calculations would indicate that for low-latitude glaciers (shown in their supplementary material) anthropogenic influence on mass balance emerges only in the last three decades. However, (a) there would be a time lag of at least a decade between the emergence of human influence on mass balance and human influence on glacier lengths (Jóhannesson, Ray- mond and Waddington, 1989), (b) the glacier's length response to changes in mass balance is strongly influenced by the glacier topography and is therefore specific to the landscape on which the glacier lies, and (c) the results of Marzeion et al. (2014), if taken at face value, would indicate that mass balance was strongly negative throughout the 20th Century. Therefore, if they were an accurate representation of Palcaraju glacier's mass balance, significant glacial retreat would be expected throughout the historical period, rather than only emerging in recent decades as has been observed (as described by the Huggel et al. manuscript). Consequently, the concurrence between the period of time for which mass loss is most attributable to anthropogenic green- house gas emissions (according to Marzeion et al., 2014) and the period of most rapid expansion of Lake Palcacocha does not appear to be a solid foundation on which to make a statement of the role of human influence on the observed retreat of this glacier.

With respect to the effect of Lake Palcacocha's expansion on the GLOF hazard, the authors explain that (lines 210-213):

"Some of the factors [influencing GLOF magnitude and probability] (such as lake formation) are closely related to climate change while others can be associated to geologic or geotechnical conditions (e.g. dam stability), or are explicitly influenced by human intervention aiming at reducing the risk of GLOFs (e.g. lake freeboard determined by the height of the constructed drainage canal)."

Anthropogenically-driven glacial retreat has directly resulted in the formation of a large proglacial lake and is the primary and necessary cause of GLOF risk. Non-climatic factors mediate the impact of climate change on the GLOF hazard, but in our view the authors should state explicitly that these factors are less important causes of the present GLOF hazard. The need to implement adaptive measures to reduce GLOF risk is therefore the result of the expansion of Lake Palcacocha and the other factors identified by the authors are largely incidental and would have been inconsequential but for the impact of climate change on the glacier's retreat.

2. Accuracy of the framework for understanding risk, and its relevance for ongoing legal proceedings

The paper gives comparable weight to the physical and 'interacting socio-economic, institutional and cultural processes' which contribute to flood risk. The existence of these social influences on flood risk is not challenged here but this framing obscures the fact that present GLOF risk is a direct result of the anthropogenically-driven retreat of the glacier.

Huggel et al present 'risk (and associated loss and damage) [as] a multi-faceted construction' and argue that 'the question of causality can often not be fully solved, at least not in a quantitative way' and 'in contrast [to the physical GLOF risk causal chain] the current conditions of exposure and vulnerability of people and values in Huaraz to GLOF hazard can only be understood with a historical perspective of social, economic, political and cultural dynamics'. It is undoubtedly true that exposure and vulnerability play a crucial role in determining the ultimate risk, however, non-climatic factors, such as the decision of Spanish colonialists to select this location for the city of Huaraz in the 16th

Century (etc) would not have mattered if not for climate change and thus are, in our view, legally irrelevant as far as the question of causality is concerned. Similarly, we believe that the possible failure of local government agencies to develop successful risk-reduction mechanisms is not relevant for understanding the primary cause of the present GLOF hazard and therefore need to implement costly adaptation measures. Climate change is a necessary cause of flood risk in this setting.

*Furthermore, based on the comments by Will Frank and Stuart-Smith et al. we also recognize the need for further clarification of the type of responsibilities we're addressing in our study. We do not focus on any legal responsibility. We added a clarification at the beginning of section 7 (Implications for responsibility and justice) where we first outline the four aspects when assigning responsibilities. We specifically add that this conceptualization of responsibility encompasses aspects of legal liability, explaining the link between the subject of responsibility and the object of responsibility, but clarify that our understanding of responsibility goes beyond the legal and liability framing. This also addresses the concerns of the two comments that the social, economic and institutional drivers of risk are irrelevant for the court case or decision. We basically agree on this (but the court decision will give us more precision in this respect) but for our concept of responsibility these risk drivers are clearly relevant.*

[revised manuscript text omitted]